# N-linked Fc glycosylation is not required for IgG-B-cell receptor function in a GC-derived B-cell line

Theresa Kissel [1,4] ✉, Veerle F. A. M. Derksen [1,4], Arthur E. H. Bentlage [2], Carolien Koeleman[3], Lise Hafkenscheid [1], Diane van der Woude [1], Manfred Wuhrer [3], Gestur Vidarsson [2] & René E. M. Toes [1] ✉

IgG secreted by B cells carry asparagine N(297)-linked glycans in the fragment crystallizable (Fc) region. Changes in Fc glycosylation are related to health or disease and are functionally relevant, as IgG without Fc glycans cannot bind to Fcɣ receptors or complement factors. However, it is currently unknown whether ɣ-heavy chain (ɣHC) glycans also influence the function of membrane-bound IgG-B-cell receptors (BCR) and thus the outcome of the B-cell immune response. Here, we show in a germinal center (GC)-derived human B-cell line that ɣHC glycans do not affect membrane expression of IgG-BCRs. Furthermore, antigen binding or other BCR-facilitated mechanisms appear unaffected, including BCR downmodulation or BCR-mediated signaling. As expected, secreted IgG lacking Fc glycosylation is unable to carry out effector functions. Together, these observations indicate that IgG-Fc glycosylation serves as a mechanism to control the effector functions of antibodies, but does not regulate the activation of IgG-switched B cells, as its absence had no apparent impact on BCR function.

Antibodies of the immunoglobulin G (IgG) isotype are crucial for immune-mediated protection against many pathogens. An important mechanism of IgG-mediated protection is the ability of IgG to bind to Fcɣ receptors (FcɣR), facilitating the activation of FcɣR-bearing cells. In addition, IgG can mediate complement activation, leading, for example, to opsonization or killing of pathogens and the release of chemoattractants, which is important for the recruitment of (innate) immune cells to the site of infection. IgG is produced by plasmablasts and plasma cells, which differentiate from B cells that recognize antigens via their B-cell antigen receptor (BCR)[1]. Similar to the BCR, IgG is a glycoprotein[2] that carries glycans linked to a conserved asparagine (N)-glycosylation consensus sequence (N-S-T) at position 297 in the $CH_2$ domain of the ɣ heavy chain (ɣHC). These N-linked glycans, often referred to as Fc glycans, are conserved between IgG subclasses. Only human IgG3 molecules carry additional glycans, which are linked to serine or threonine residues in the hinge region, so-called O-linked glycans[3]. Although the N-linked glycosylation of human BCRs is ill-defined, the Fc glycans expressed by IgG antibodies are well-defined. Structurally, they are mainly di-antennary (A2) complex-type glycans bearing a core fucose, varying amounts of terminal galactoses and, to a lesser extent, bisecting *N*-acetylglucosamines (attached to the core β-mannose), and terminal sialic acids[4,5].

To date, it has been extensively studied and shown that IgG-Fc glycan composition is highly variable and changes with age, sex, health, and disease[4,6,7]. Specific IgG-Fc glycosylation patterns lacking terminal galactoses have been identified, for example, in rheumatic (e.g., rheumatoid arthritis, systemic lupus erythematosus, and ANCA-associated vasculitis) and other inflammatory diseases and are clearly

[1]Department of Rheumatology, Leiden University Medical Center, 2333 ZA Leiden, The Netherlands. [2]Department of Experimental Immunohematology, Sanquin Research and Landsteiner Laboratory, Amsterdam University Medical Center, University of Amsterdam, 1006 AD Amsterdam, The Netherlands. [3]Center for Proteomics and Metabolomics, Leiden University Medical Center, 2333 ZA Leiden, The Netherlands. [4]These authors contributed equally: Theresa Kissel, Veerle F. A. M. Derksen. ✉e-mail: T.Kissel@lumc.nl; R.E.M.Toes@lumc.nl

associated with inflammation and disease activity[8]. In addition, alterations in Fc glycosylation have recently been highlighted in studies of anti-spike protein IgG responses in patients with COVID-19, showing highly dynamic glycosylation patterns, including low core fucosylation that correlates with disease severity[9–11]. Reduced core fucosylation of IgG has also been reported for alloimmune responses against cellular blood groups and for responses against glycoproteins of HIV and dengue viruses, leading to the hypothesis that membrane-embedded antigens induce a specific afucosylated B-cell response, followed by secretion of afucosylated IgG[9].

Functionally, Fc glycans have a substantial impact on the structure of soluble IgG molecules and are essential for the recruitment of effector functions such as complement activation and binding to FcγRs. The core fucose of N(297)-linked glycans on IgG has been shown to sterically collide with the fucose of N(162)-linked glycans on FcγRIII, thereby modulating antibody-dependent cellular cytotoxicity (ADCC)[12]. This explains the potent immune responses of afucosylated IgG described above, as they are able to efficiently bind and activate immune cells carrying FcγRIII[9]. At the functional level, changes in Fc galactosylation have been shown to enhance complement activation, as hypergalactosylated IgG-Fc domains have a higher potential for hexamerization, allowing more efficient interaction with the initiator of the classical complement pathway C1q, thereby enhancing complement-dependent cytotoxicity (CDC)[13,14].

Fc glycosylation is thus essential for the function of IgG secreted by B cells, but the role of γHC carbohydrates in BCR function remains unclear. Since B-cell development, survival, and activation depend on the expression of functional BCRs[15], it is important to investigate whether Fc glycans influence their structure and function. The IgG-BCR consists of an immunoglobulin (Ig)-like structure that resembles the structure of secreted IgG molecules, but is membrane-bound (mIgG) and thus extended by long transmembrane helices. In addition, the IgG-BCR forms a non-covalent complex with the signal-transducing heterodimer Igα/Igβ (CD79α and CD79β) at a 1:1 stoichiometry[16,17]. After antigen engagement, the intracellular immunoreceptor tyrosine-based activation motifs (ITAM) in the Igα/Igβ subunits get phosphorylated by the kinase Lyn, which then triggers a cascade of signaling events, resulting in B-cell activation, differentiation, and antibody production[18].

That glycosylation can alter the function of cellular immune receptors has already been shown for, e.g., CD22[19,20], MHC-II[21], T-cell receptors[22], and IgM-BCRs. In particular, a study using mouse pre-B-cell lines has shown that N(46)-linked glycans in the CH$_1$ domain of μHC are required for IgM pre-BCR formation and function[23]. However, in contrast to IgG-BCRs, membrane-bound IgM (mIgM) BCRs are heavily glycosylated and contain a total of four conserved N-glycosylation sites. In addition, there are notable differences in the assembly and positioning of IgG- and IgM-BCRs on the cell surface. Compared with mIgM, mIgG contains cytoplasmic domains of considerable length, including signal-amplifying peptide motifs, e.g., the immunoglobulin tail tyrosine (ITT) motif, which plays a key role in IgG-B cell activation[24]. In addition, recent cryo-EM studies have shown that the Fc domain of mIgM is located closer to the plasma membrane than that of mIgG[16,25]. These alterations could result in different roles of N-glycosylation for different BCR isotypes. For IgG-BCRs, a study knocking out fucosyltransferase (Fut8) in B cells suggests that glycosylation is important for IgG-BCR function[26], although it is unclear whether the observed effects are due to fucosylation changes in the BCR or by prevention of core fucose incorporation on N-linked glycans of other cellular proteins[27].

Given the importance of γHC glycosylation for IgG antibody function, we have now developed tools to investigate the effects of glycosylation on BCRs by generating a γHC glycan-site mutant in human germinal center (GC)-derived B cells from Burkitt Lymphoma (Ramos). Intriguingly, we show that Fc glycosylation has no effect on IgG-BCR function, including antigen recognition, signal transduction,

BCR downmodulation, concomitant antigen uptake, and antibody secretion. Our results indicate that γHC glycans do not regulate BCR-mediated humoral immune responses but have evolved to direct the effector functions of antibodies produced by these B cells.

## Results

### Lack of γHC glycosylation has no effect on IgG-BCR surface expression

We generated human B-cell lines from Burkitt Lymphoma (Ramos) expressing IgG-BCRs in the presence or absence of γHC glycosylation. To this end, Ramos B cells were knocked-out (KO) for their endogenous IgM and IgD BCR and the activation-induced cytidine deaminase (AID) enzyme (referred to as MDL-AID KO), as previously described[28,29]. B-cell lines were generated by transducing the BCR-negative MDL-AID KO cell line with N(297)-γHC or mutated Q(297)-γHC IgG-BCR sequences and the transduction marker GFP (Fig. 1a). The BCR sequences used were obtained from single-cell sorted human B cells isolated from patients with the autoimmune disease rheumatoid arthritis as previously described[28]. Two BCRs were directed towards citrullinated antigens (2G9 and 3F3), while a third BCR was directed against tetanus toxoid (D2). We next compared the surface expression of the IgG-BCRs in the presence or absence of their Fc glycans (FcG). All three GFP + B-cell lines (2G9, 3F3 and D2) were able to stably express membrane-bound IgG (mIgG) BCRs in the absence of FcG over a time period of 20 days (Fig. 1b–d; Supplementary Fig. 1a–c). Next, we captured IgG-BCRs and identified the expected apparent molecular weight, indicating a structurally intact BCR (Fig. 1e). The FcG−negative (-) mIgG displayed a smaller size compared to the FcG−positive (+) mIgG, as expected due to the absence of two glycans in the γHC. Since few data are available on glycans expressed by human BCRs, we aimed to analyze the Fc glycans after IgG-BCR capture and tryptic digestion by liquid chromatography coupled with mass spectrometry (LC-MS). As expected, no Fc peptide glycans could be detected for the FcG− B-cell lines (Fig. 2a, c). The glycans expressed in the non-mutated γHC of the IgG-BCRs were mainly di-antennary (A2) complex-type glycans containing core fucosylation, 77% (2G9 IgG-BCR), 88% (3F3 IgG-BCR) or 78% (D2 IgG-BCR) galactosylation, 47% (2G9 IgG-BCR), 52% (3F3 IgG-BCR) or 46% (D2 IgG-BCR) bisecting N-acetylglucosamines and 16%, (2G9 IgG-BCR), 21% (3F3 IgG-BCR) or 19% (D2 IgG-BCR) terminal sialic acids (mainly S0 or S1 glycans) (Fig. 2a–c; Supplementary Fig. 2a, b). The expression of complex-type N(297)-glycans on mIgG was confirmed by cell surface biotinylation and subsequent Western blot analysis of biotinylated IgG treated with EndoH (cleaving only high-mannose structures) or PNGaseF (cleaving all N-glycan structures) (Supplementary Fig. 2c). γHC N-glycans on mIgG could only be cleaved after PNGaseF treatment, as indicated by a size-shift, supporting the absence of high-mannose glycans on these mIgG.

Together, these results indicate that γHC glycosylation has no effect on IgG-BCR assembly, cellular trafficking, and its membrane expression.

### IgG-BCR Fc glycans do not impact antigen binding or BCR downmodulation

To test whether γHC glycosylation of IgG-BCRs affects binding to antigens, we determined the binding of the three FcG+ and FcG− mIgG B-cell lines to their respective antigens by flow cytometry. The two B-cell lines (2G9 and 3F3) directed against citrullinated antigens [cyclic-citrullinated peptide 2 (CCP2)] showed binding to the citrulline but not to the unmodified arginine-containing control peptide (CArgP2) (Fig. 3a). The D2 B-cell line showed binding to the tetanus toxoid (TT) antigen, as expected. No binding was observed for the MDL-AID KO control cell line, which

lacks IgG-BCRs. The absence of Fc glycosylation had no effect on the binding of the IgG-BCRs to their respective antigen (Fig. 3a), also at lower antigen concentrations (Fig. 3b). Next, we performed a flow-based assay to investigate the downmodulation of IgG-BCRs after (antigenic) stimulation and to determine the effects of FcG on BCR internalization and concomitant antigen uptake. To this end, B-cell lines were stimulated with citrullinated peptide (CCP2)-tetramers or TT-tetramers on ice, followed by incubation at 37 °C for several minutes to allow uptake of the antigen-bound IgG-BCR complex. Subsequently, the expression of the remaining BCRs on the B-cell surface was detected (Supplementary Fig. 3). Our data show no effect of FcG on IgG-BCR and concurrent antigen uptake after 5, 15, or 30 min incubation at 37 °C (Fig. 3c). No BCR downmodulation was observed after

stimulation with the arginine control peptide (CArgP2). Similarly, when using TT, we observed no effect of FcG on TT-specific IgG-BCR uptake. To substantiate these data, we also employed conventional techniques to stimulate and cross-link the BCR using Fab'2 directed against the kappa light chain constant domain to achieve robust downmodulation of the BCR. Again, FcG showed no effect on IgG-BCR internalization (Fig. 3c).

These data demonstrate that γHC glycosylation is not required for antigen-bound IgG-BCR uptake.

### IgG-BCR Fc glycosylation has no effect on BCR signal transduction

Based on the close proximity of the mIgG N(297)-linked glycans and the transmembrane Igα/Igβ signaling complex[16], we

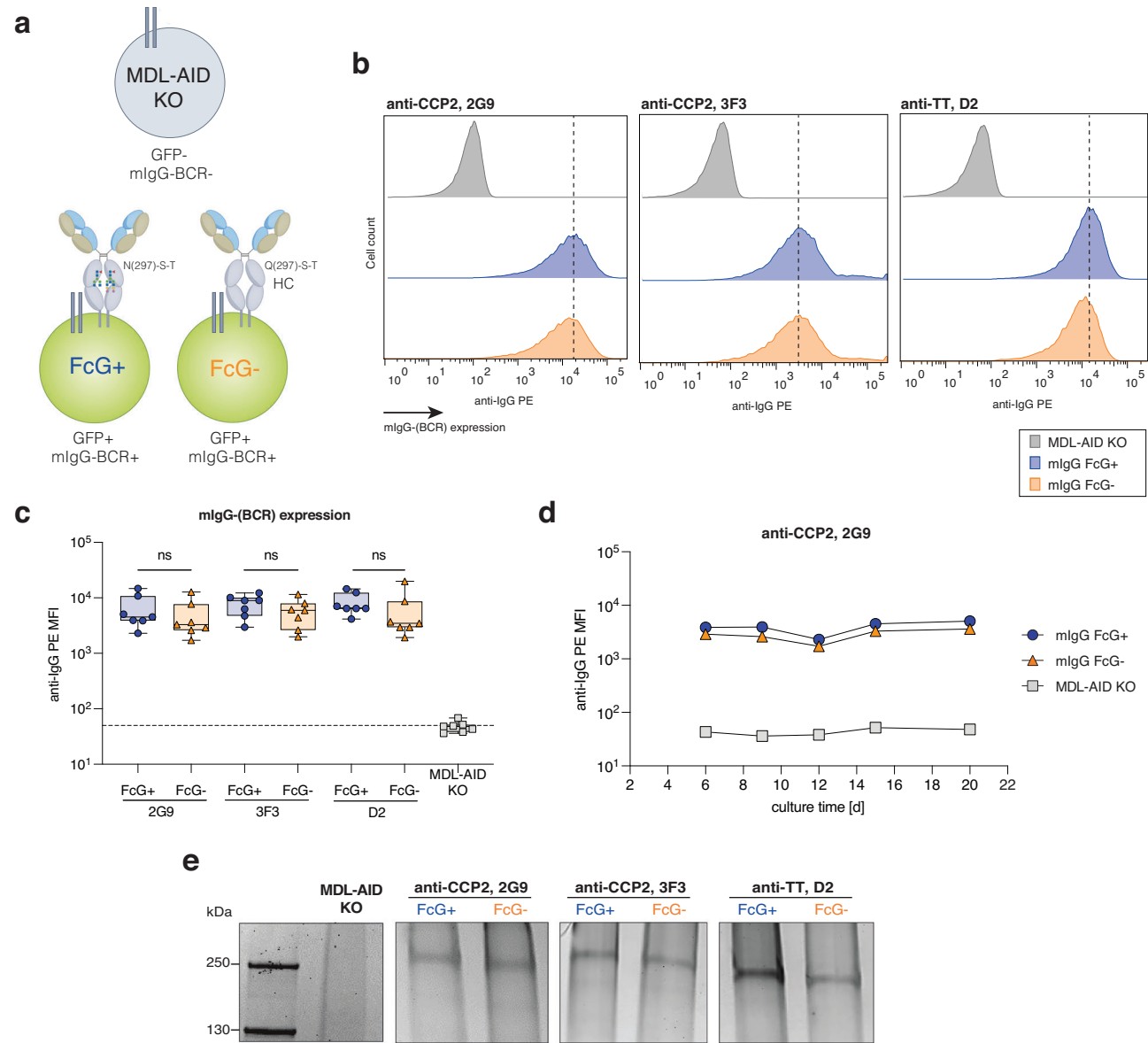

**Fig. 1 | Membrane-bound IgG (mIgG) [B-cell receptor (BCR)] expression in the presence and absence of Fc glycans (FcG). a** Schematic representation of generated Burkitt Lymphoma (Ramos) B-cell lines. B cells knocked out for their endogenous BCR and AID (MDL-AID KO), and B cells transduced with GFP and mIgG-BCR sequences, including the conserved glycosylation site N(297)-S-T or mutant Q(297)-S-T to remove Fc glycans (FcG). **b** Flow histograms and **c** bar graphs of mIgG-BCR expression in the presence (FcG+) or absence of Fc glycans (FcG−). No BCR surface expression on MDL-AID KO cell line. Box plots show median,

interquartile values, range, and all individual data points. *n* = 7 biologically independent experiments. ns (not significant) *p* > 0.05 (two-sided unpaired *t* tests). **d** mIgG-(BCR) expression of anti-CCP2 (2G9) FcG+ and FcG− and MDL-AID KO B cells over culture time (14 days). **e** SDS-PAGE of captured mIgG from FcG+ and FcG− B cell lines, two anti-CCP2 (2G9 and 3F3) and one anti-TT (D2), and MDL-AID KO control cells. Representative results of 2 biological replicates are shown. Source data are provided as a Source Data file.

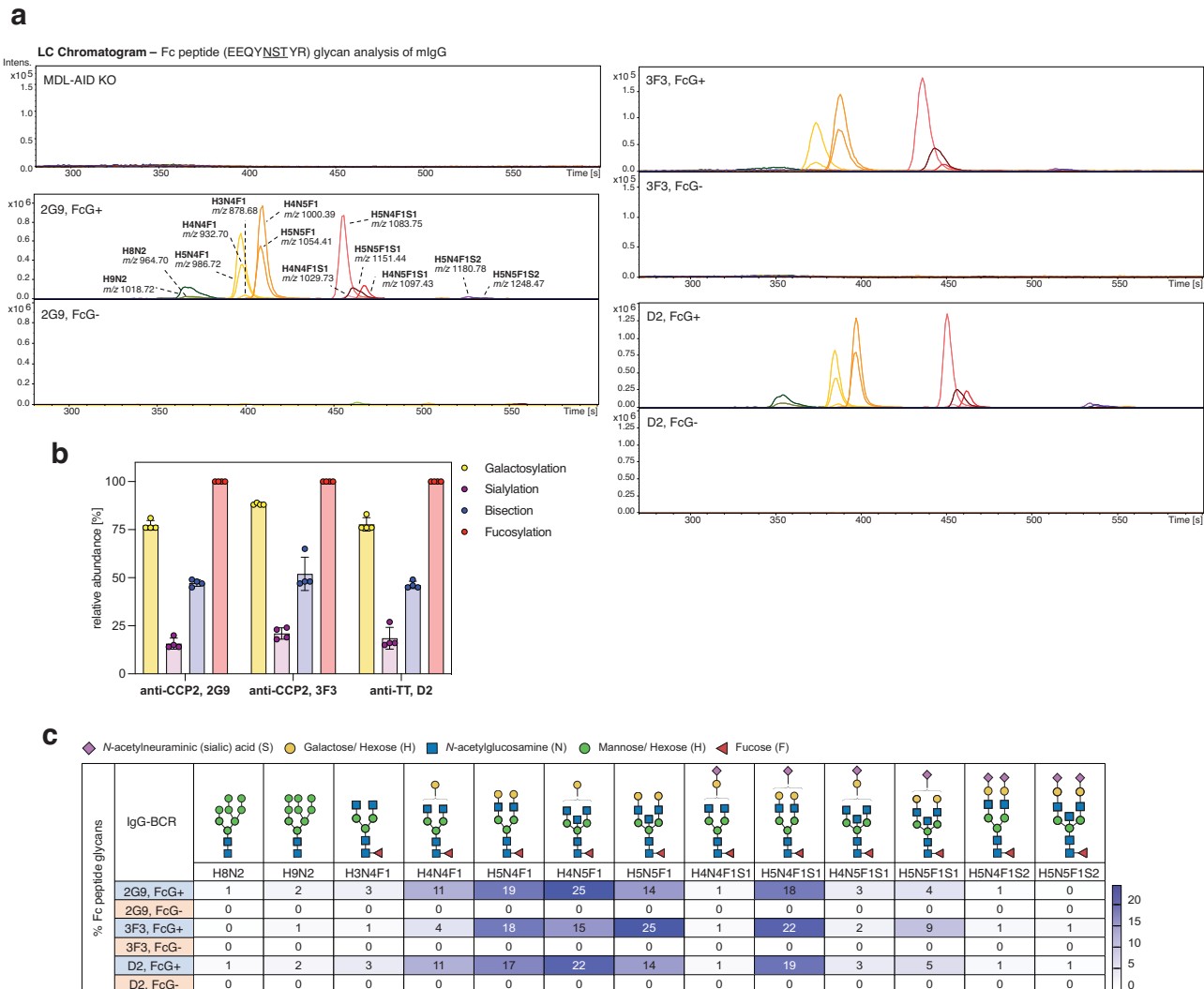

**Fig. 2 | Fc peptide glycan analysis of mIgG-BCRs. a** LC-MS Fc peptide glycan analysis of captured FcG+ and FcG− mIgG of all B-cell lines (2G9, 3F3, and D2). LC chromatograms of glycan peaks (with annotation and *m/z*) are shown. **b** Percentage of galactosylation, sialylation, bisection, and fucosylation on mIgG γHC glycans (FcG+ B-cell lines). Bar graphs show mean, standard error, and individual data points. *n* = 4 biologically independent experiments. **c** Percentage of individual γHC glycan traits expressed on 2G9, 3F3, and D2 mIgG-BCRs. Glycan traits are schematically depicted. Heat map shows *n* = 4 biologically independent experiments. Source data are provided as a Source Data file. H Hexose, N N-acetylglucosamine, F fucose, S sialic acid.

hypothesized that FcG might affect signal transduction of IgG-BCRs. We investigated the activation of the IgG-BCR signaling complex by measuring phosphorylation of Syk kinase (Supplementary Fig. 4a) and calcium flux (Supplementary Fig. 4c) following BCR stimulation. BCRs were stimulated either with their respective antigen (CCP2 and TT) or Fab'2 antibodies directed against the constant domain of mIgG. All B-cell lines showed increased phosphorylation of Syk (pSyk) after stimulation compared to the PBS treatment control (Fig. 4a, b; Supplementary Fig. 4b). We did not detect any effect of γHC glycosylation on the activation of the IgG-BCR complex, as indicated by similar median fluorescence pSyk intensities between the FcG− and FcG+ B-cell lines (Fig. 4a, b; Supplementary Fig. 4b). Similar pSyk expression was also observed after stimulating with titrated amounts of anti-IgG or antigen or when stimulating for different time points, further indicating that γHC glycosylation does not affect Syk-mediated BCR signaling or feedback signaling mechanisms (Fig. 4c, d; Supplementary Fig. 4d). Consistent with these results, calcium flux peak levels (Fig. 5a, b) and kinetics (Fig. 5c; Supplementary Fig. 4e) were also similar between FcG− and FcG+ mIgG

B-cell lines while no calcium flux was observed after stimulation with PBS (Fig. 5a).

Thus, the results are consistent with the data on BCR down-modulation and indicate that N(297)-linked glycans in the Fc domain of mIgG do not affect the function of the BCR:Igα/Igβ signaling complex.

## Human B-cell lines can produce IgG in the absence of γHC glycosylation

To determine whether Fc glycosylation affects IgG secretion by B cells, we next transduced the MDL-AID KO Ramos B cells with N(297)-γHC or mutant Q(297)-γHC IgG sequences in the absence of the transmembrane domain (Fig. 6a). We then screened the supernatant of Ramos B cells for the presence of FcG− or FcG+ 2G9 IgG with an antigen-specific (CCP2) IgG ELISA. The transduced Ramos B cells were able to secrete an average of ~550 ng/ml of 2G9 IgG, while the non-transduced MDL-AID KO cells did not. We could not detect an effect of FcG on 2G9 antibody secretion (Fig. 6b). Furthermore, the secreted antibodies had the expected apparent molecular weight as determined by gel electrophoresis, with a marginal size-shift between the FcG− and FcG+

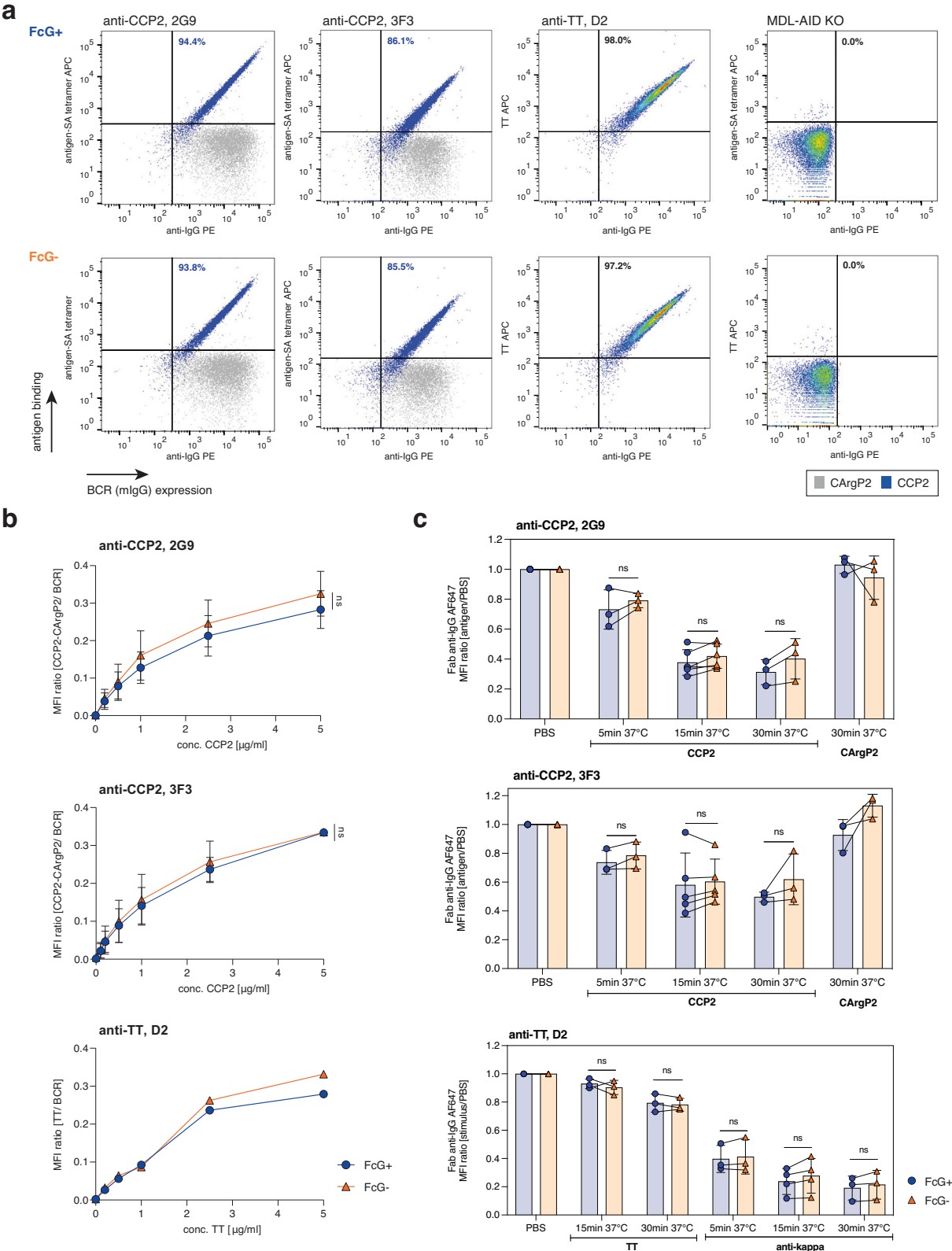

slightly lower bisection (39%), and terminal sialic acid (12%) levels than 2G9 mIgG (Fig. 6f).

These data demonstrate that human B cells are capable of secreting IgG in the absence of γHC glycosylation and that sIgG and mIgG express similar complex-type A2 glycans at N(297) characterized by core fucosylation, terminal galactosylation, various amounts of bisection and the presence of one sialylated antennae. The data also

variants, due to the absence of two γHC glycans (Fig. 6c). We confirmed the absence of the γHC glycans in the FcG− secreted IgG (sIgG) by Fc peptide glycan analysis and LC-MS (Fig. 6d, e). Comparable glycan profiles were obtained for the FcG+ sIgG and mIgG (Supplementary Fig. 2d). Analogous to 2G9 mIgG, the Fc glycans on 2G9 sIgG were mainly S0 or S1 complex-type glycans with a core fucose. 2G9 sIgG had similar galactosylation (79%),

**Fig. 3 | mIgG-BCR FcG does not affect antigen binding and subsequent BCR downmodulation. a** Gating strategy to assess BCR (mIgG) expression and antigen binding of human Ramos B-cell lines. Binding to the cyclic-citrullinated peptide 2 (CCP2, blue) and the unmodified arginine control peptide (CArgP2, gray) was assessed for 2G9 and 3F3 (overlay dot plot is shown), and binding to tetanus toxoid (TT) for the D2 B-cell line. No antigen binding was observed for the MDL-AID KO B cells that lack BCRs. **b** Antigen binding titration curves of 2G9, 3F3, and D2 BCRs to their respective antigens (CCP2-CArgP2 or TT) in a concentration range of 0−5 μg/ml. The median fluorescence intensity (MFI) relative to the BCR expression is shown. Points show mean and standard error. $n = 3$ biologically independent experiments (2G9 and 3F3) or $n = 2$ biologically independent experiments (D2). ns

(not significant) $p > 0.05$ (two-sided multiple paired $t$ tests). **c** Downmodulation of 2G9, 3F3, and D2 mIgG-BCRs after PBS, antigen, or anti-kappa stimulation and incubation at 37 °C for 5, 15, and 30 min. The MFI of the remaining mIgG-BCRs detected using Fab anti-IgG is depicted (ratio between MFI of stimulated and PBS-treated cells). Bar graphs show mean, standard error, and paired individual data points. $n = 3$ biologically independent experiments (5 min and 30 min CCP2/CArgP2/TT or anti-kappa stim. and 15 min TT stim.), $n = 4$ biologically independent experiments (15 min anti-kappa stim.), $n = 5$ biologically independent experiments (15 min CCP2 stim. of 3F3) or $n = 6$ biologically independent experiments (15 min CCP2 stim. of 2G9). ns $p > 0.05$ (two-sided paired $t$ tests). Source data are provided as a Source Data file.

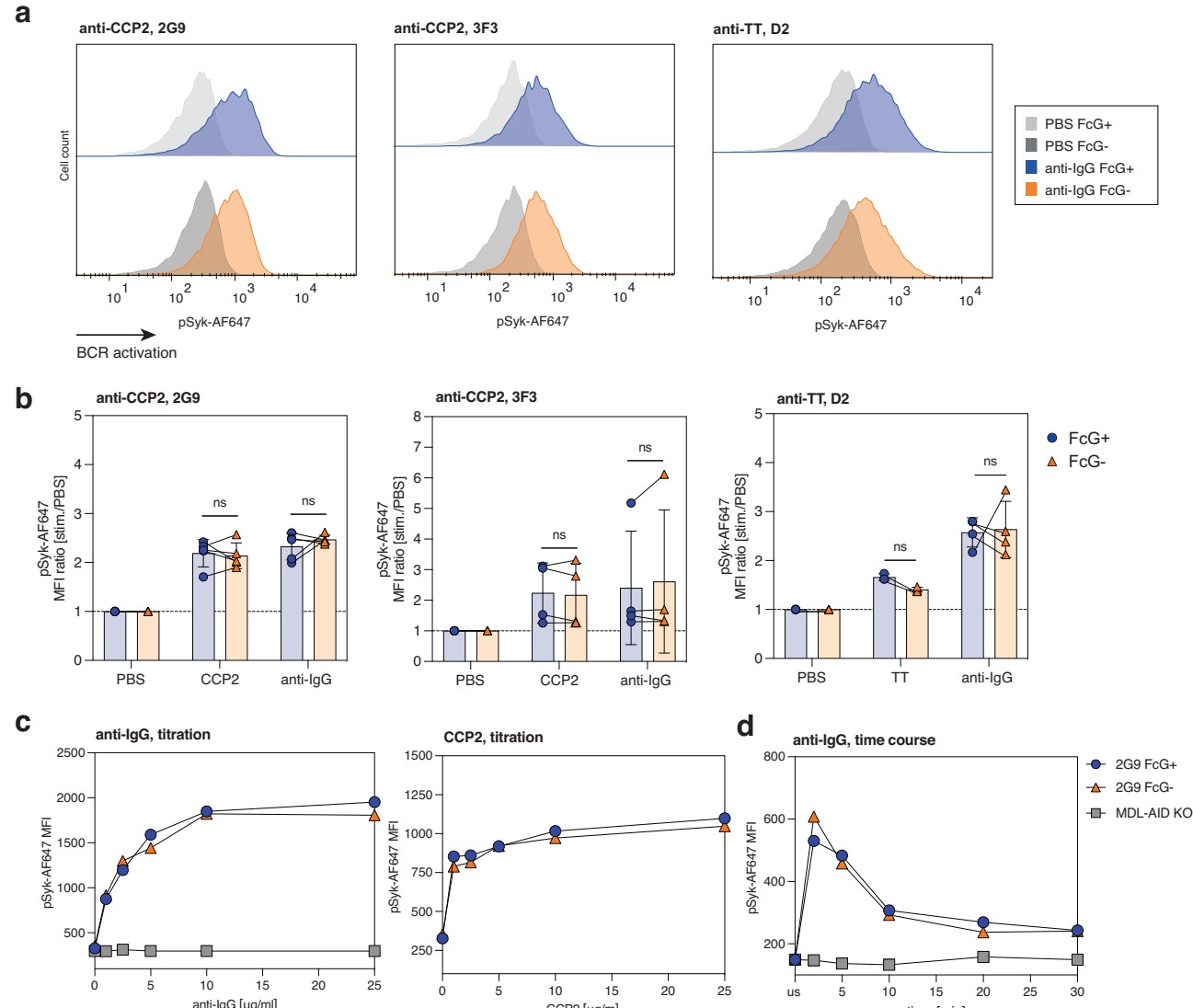

**Fig. 4 | No influence of mIgG FcG on BCR signal transduction. a** Flow histograms showing phosphorylation of Syk (pSyk) of 2G9, 3F3, and D2 FcG+ and FcG− B cells after 5 min PBS and anti-IgG stimulation (5 μg/ml). **b** pSyk MFI values of 2G9, 3F3 and D2 FcG+ and FcG− B cells after 5 min stimulation. The ratio towards PBS unstimulated cells is depicted. Bar graphs show mean, standard error, and paired individual data points. $n = 3$ biologically independent experiments (5 μg/ml TT stim. of D2), $n = 4$ biologically independent experiments (5 μg/ml CCP2 stim. of 3F3 and 5 μg/ml anti-IgG stim of 3F3/D2) or $n = 5$ biologically independent experiments

(C5 μg/ml CP2 and 5 μg/ml anti-IgG stim. of 2G9). ns (not significant) $p > 0.05$ (two-sided paired $t$ tests). **c** pSyk MFI values of 2G9 FcG+ and FcG− B cells after 5 min stimulation with 0−25 μg/ml anti-IgG and CCP2. Representative results of two biological replicates are shown. **d** pSyk MFI values of 2G9 FcG+ and FcG− B cells unstimulated (us) and after 5−30 min stimulation with 5 μg/ml anti-IgG. MDL-AID KO cells are shown as control. Representative results of two biological replicates are shown. Source data are provided as a Source Data file.

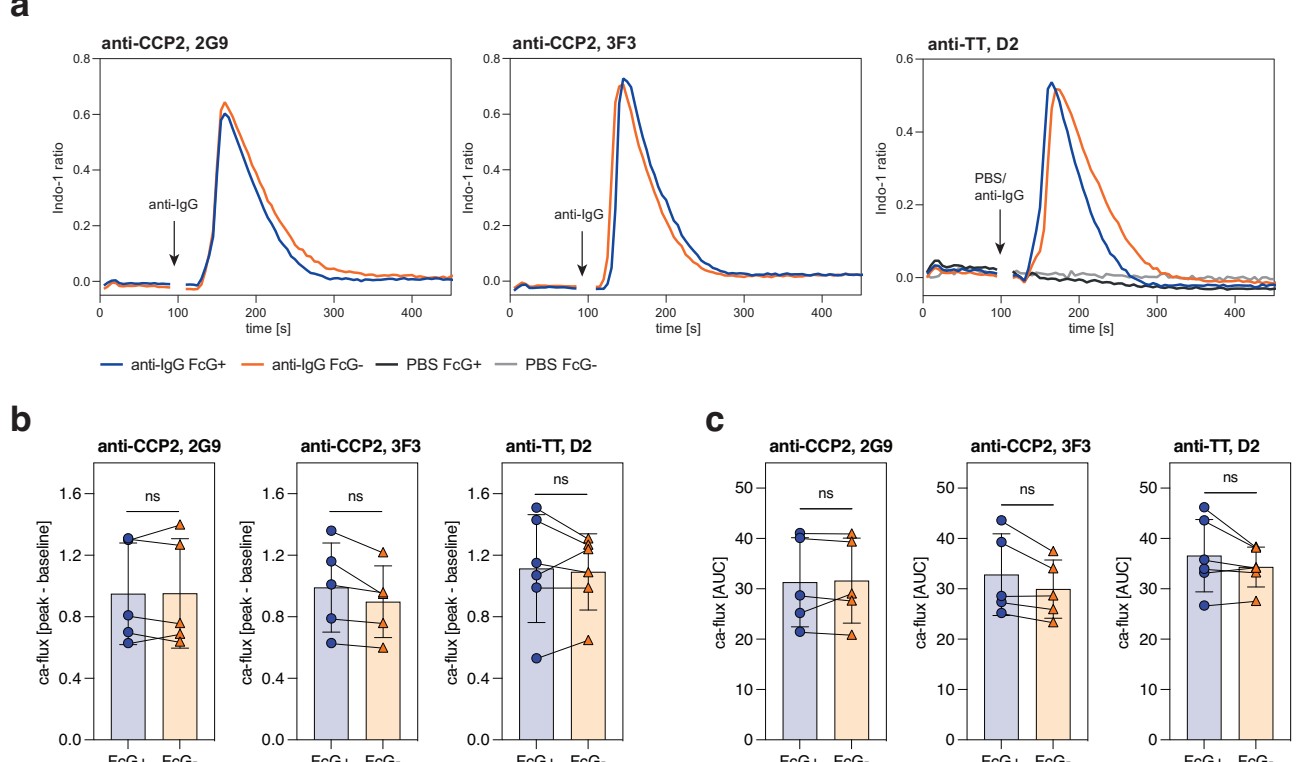

**Fig. 5 | mIgG FcG does not affect BCR-mediated calcium flux. a** Calcium flux (Calcium-bound Indo-1/unbound Indo-1) histograms of 2G9, 3F3 and D2 FcG+ and FcG− B cells after anti-IgG stimulation. PBS treatment control is shown for D2 cell lines. **b** Calcium flux (peak-baseline) and **c** kinetics (area under the curve, AUC) of FcG+ and FcG− B-cell lines. Bar graphs show mean, standard error, and paired individual data points. $n = 5$ biologically independent experiments (2G9 and 3F3) or $n = 6$ biologically independent experiments (D2). ns (not significant) $p > 0.05$ (two-sided paired $t$ tests). Source data are provided as a Source Data file.

indicate that cellular trafficking is not affected by FcG and that sIgG and mIgG follow similar biosynthesis pathways.

### Secreted IgG are unable to recruit effector functions in the absence of Fc glycans

Our results suggest that FcG is not required for mIgG-BCR function. To rule out the possibility that the IgG used in this study are unusual in their biological properties, we next wanted to confirm that their ability to recruit effector functions depends on the presence of FcG.

First, we tested the sIgG for their ability to bind human FcγR using surface plasmon resonance (SPR). As expected, FcG− sIgG were unable to bind to human FcγRII/IIIa and b and showed only low binding to the high-affinity FcγRI ($K_D = 250$ nM) (Fig. 7a, b). In contrast, the FcG+ sIgG showed binding to all assessed FcγRs. Affinities could be calculated for the medium-affinity receptors FcγRIIa 131H ($K_D = 433$ nM), FcγRIIa 131 R ($K_D = 507$ nM), FcγRIIIa 158 F ($K_D = 573$ nM), FcγRIIIa 158 V ($K_D = 362$ nM) and for the high-affinity FcγRI ($K_D = 12$ nM) (Fig. 7b), whereas no affinity calculation could be performed for binding to the low-affinity FcγRIIb and IIIb NA2 (Supplementary Fig. 5). In addition to FcγR binding, we determined the effect of FcG on activation of the classical complement pathway using a previously described and established complement ELISA[30]. Both sIgG were able to bind to the antigen (CCP2)-coated ELISA plate, indicating the production of intact IgG molecules and supporting the finding that FcG does not affect antigen binding (Fig. 7c). However, we could only observe the binding of C1q and subsequent C4 and C3c deposition to the antigen-bound IgG that expressed γHC glycans (Fig. 7d). The Q(297)-γHC mutant showed complement deposition ELISA signal intensities that were comparable to signals observed in the absence of exogenous complement (NHS) (Fig. 7d).

In summary, our results show that γHC glycosylation is crucial for sIgG to exert its effector functions, such as binding to FcγRs and activation of the classical complement cascade.

## Discussion

Here, we have shown that while γHC glycosylation is indispensable for secreted IgG to recruit effector functions, it is paradoxically not required for the function of membrane-bound IgG-BCRs. Interestingly, no effect of FcG on a stable BCR surface expression and B-cell activation was observed, although the N(297)-linked glycans on mIgG are in close proximity to the ITAM-bearing Igα/Igβ signaling complex and are involved in shaping the 3D-structure of BCRs[16]. BCR expression was measured over a 20-day period to exclude the possibility that fluctuations in mIgG surface expression influenced subsequent functional responses. Furthermore, we have shown that neither antigen binding nor subsequent BCR-antigen uptake is affected by γHC glycosylation and that human B cells secrete aglycosylated antibodies in a similar manner as their glycosylated counterparts. However, when secreted IgG lacks FcG, it can no longer recruit effector functions.

A previous study reported that core fucosylation of mIgG is required for BCR function[26]. However, it is conceivable that the experimental design used by the authors is not suitable to address this critical question, as core fucosylation was not studied in a BCR-specific manner, but rather ubiquitously for all N-linked glycans. It is, therefore, likely that an aberrant B cell N-glycome affects B-cell function rather than BCR Fc glycosylation as described.

To investigate the effects of IgG-BCR Fc glycans on B-cell function, we made use of the Burkitt lymphoma B-cell line (Ramos) and knocked out the N(297)-linked glycosylation site on mIgG. Ramos cells are human GC-derived B cells that express co-receptors in close proximity

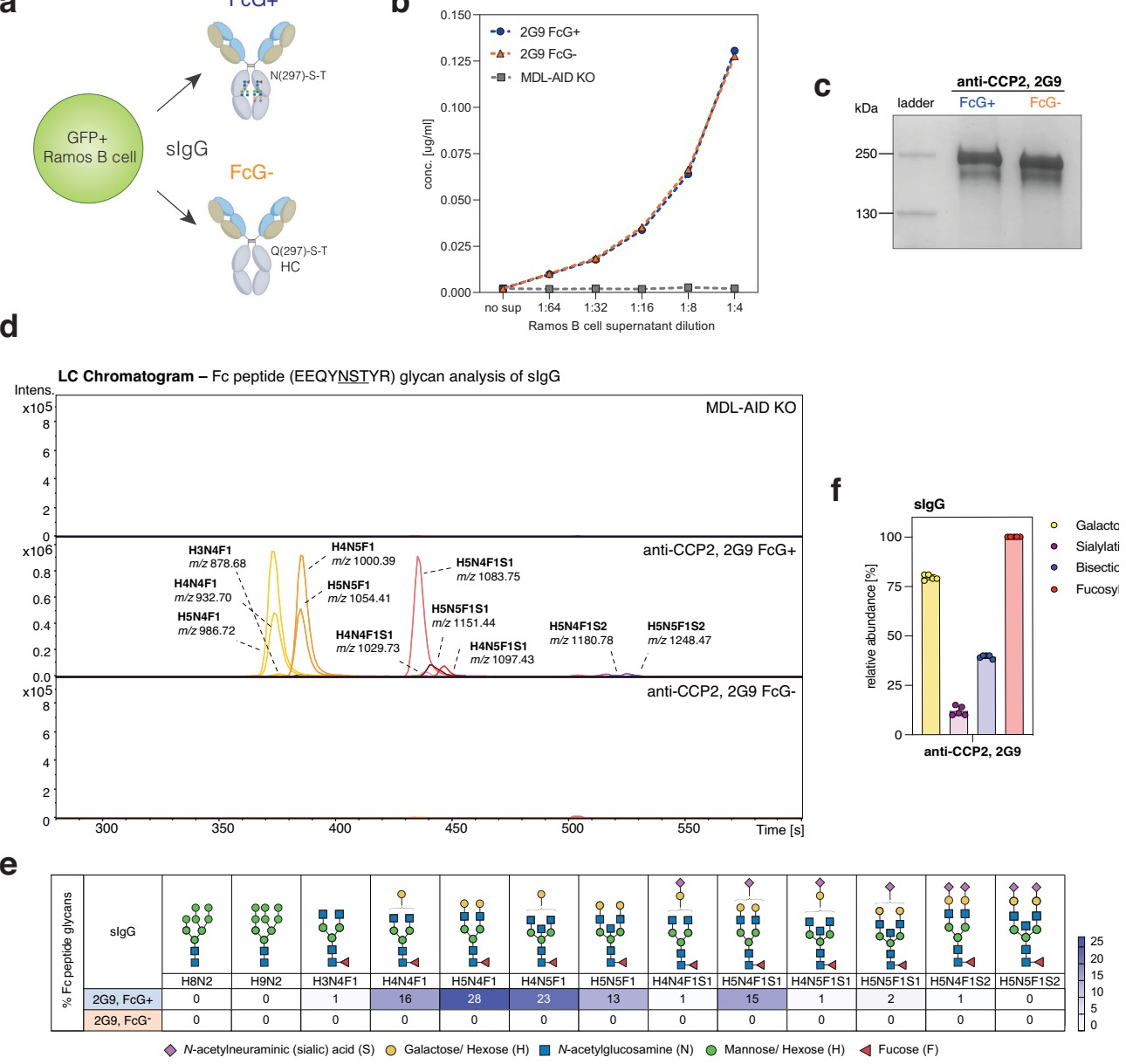

**Fig. 6 | B-cell IgG secretion in the presence and absence of γHC glycosylation.**
**a** Schematic representation of Ramos B cells secreting IgG with either N(297)-S-T or mutant Q(297)-S-T γHCs. **b** 2G9 IgG FcG+ and FcG− secreted by Ramos B cells detected using an antigen-specific (CCP2) IgG ELISA. No anti-CCP2 IgG was detected in B-cell supernatant of MDL-AID KO cells that don't secrete IgG. The mean and standard error of 2 technical replicates (one experiment) are shown. **c** SDS-PAGE of Ramos B cell secreted 2G9 IgG with and without FcG. Representative results of 3 biological replicates are shown. **d** LC-MS Fc peptide glycan analysis of captured FcG

+ and FcG− 2G9 sIgG. LC chromatograms of glycan peaks (with annotation and *m/z*) are shown. **e** Percentage of γHC glycan traits expressed on 2G9 sIgG. Glycan traits are schematically depicted. Heat map shows mean of 4 biological replicates.
**f** Percentage of galactosylation, sialylation, bisection, and fucosylation on mIgG γHC glycans (FcG+ B-cell lines). Bar graphs show mean, standard error, and individual data points. *n* = 5 biologically independent experiments. Source data are provided as a Source Data file. H Hexose, N *N*-acetylglucosamine, F Fucose, S sialic acid.

to the BCR that can regulate B-cell fate, such as CD19, CD20, and the sialic acid binding immunoglobulin-like lectin CD22 (Siglec-2)[31,32]. We hypothesized that FcG could affect the co-localization of these regulatory receptors with the BCR and thus influence B-cell activation. For example, there is evidence suggesting that sialic acids expressed on IgM-BCR N-glycans interact with the lectin CD22 to regulate downstream signaling events[33]. It is, therefore, tempting to speculate that sialylated IgG-BCR N-glycans also interact with CD22. However, we show here that cross-linking between the IgG-BCR and CD22 via glycan-lectin interactions is unlikely, as we did not observe any effect of FcG knock-outs on BCR signaling events. This could likely be explained by the differences in mIgG- vs mIgM-BCR assembly and

positioning[16,24,25]. In addition, the long cytoplasmic tail of mIgG, which amplifies signaling independent of the Igα/Igβ signaling complex[24], could potentially limit the effect of γHC glycans on B-cell activation.

Nevertheless, we cannot exclude the possibility of interactions with other lectins expressed on the B cell surface (in *cis*) or on surrounding immune cells (in *trans*) that are not present in our B-cell culture system. However, we consider *trans* interactions less likely because FcG on IgG-BCRs are "hidden" between the two γHCs and are therefore less accessible to *trans*-lectins than, for example, glycans expressed in the variable domains. In addition, similar to IgM-BCRs[34], IgG-BCR Fc glycans could modulate the spatial organization of the BCR relative to other co-receptors through interactions with secreted

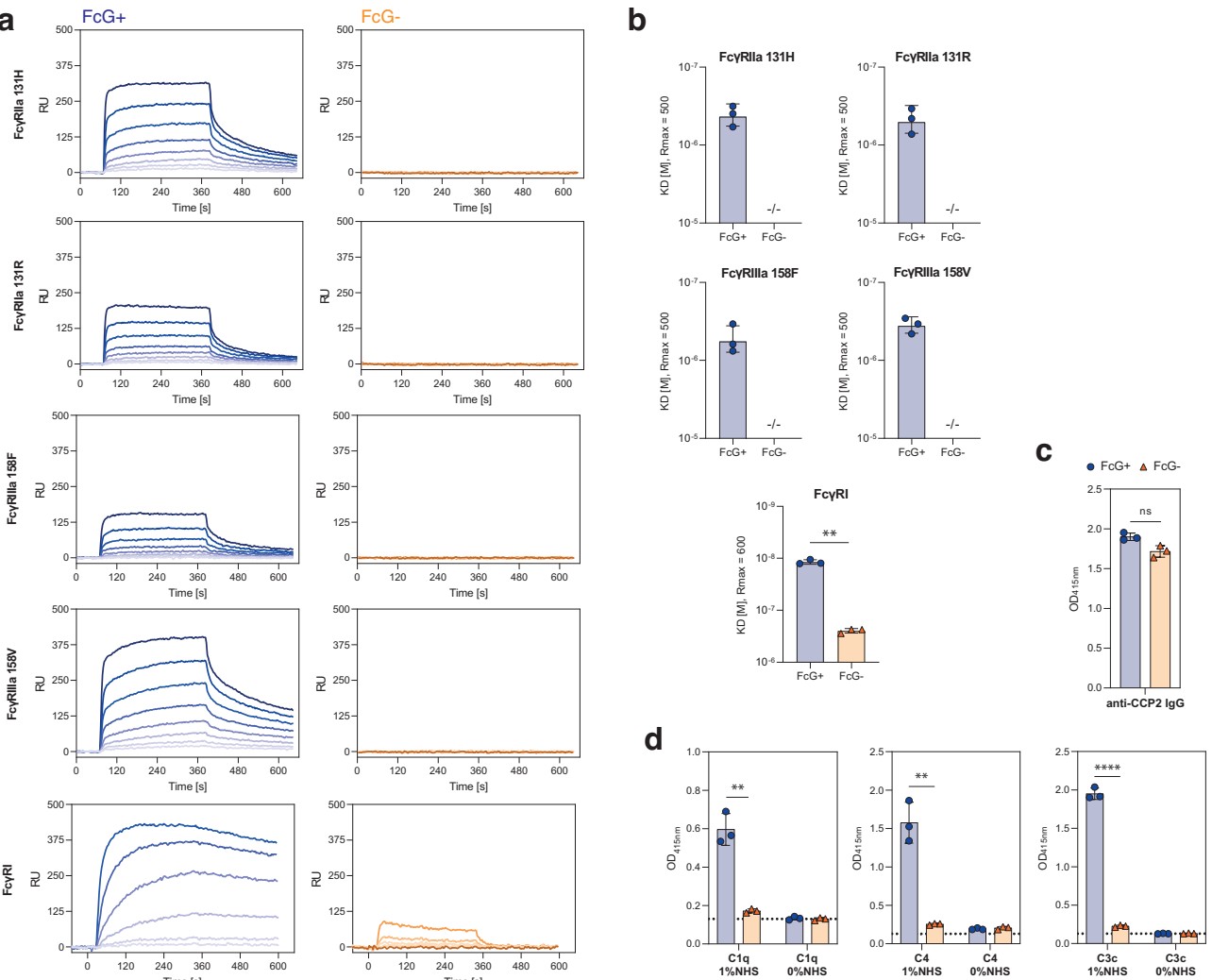

**Fig. 7 | Impact of IgG FcG on the recruitment of effector functions.**
**a** Representative SPR sensorgrams of 2G9 sIgG FcG+ and FcG− binding to human FcγRI and II/IIIa. Association and dissociation are represented as response units [RU] over time [s]. **b** $K_D$ values of binding of FcG+ and FcG− 2G9 sIgG to human FcγRI and II/IIIa as determined by SPR. Bar graphs show mean, standard error, and individual data points. $n = 3$ biologically independent experiments. **$p = 0.0036$ (two-sided paired $t$ test). **c** Detection of intact antigen-specific (CCP2) IgG and

**d** antigen-specific classical complement activation of FcG+ and FcG− 2G9 sIgG determined by ELISA. Deposition of complement components C1q, C4, and C3c was detected in the presence or absence of exogenous complement (normal human serum, NHS). Bar graphs show mean, standard error, and individual data points. $n = 3$ technical replicates of one experiment. ns (not significant) $p > 0.05$, **$p = 0.001$ (C4), **$p = 0.0009$ (C1q), ****$p = 0.000002$ (C3c) (two-sided unpaired $t$ tests). Source data are provided as a Source Data file.

lectins, e.g., galectin-9, thereby influencing B-cell activation. The possible effect of Fc glycans on the spatial distribution of IgG-BCRs could potentially also apply to core fucosylation, thereby explaining some of the in vivo results described previously[26]. Thus, although our data do not support a direct influence of Fc glycans, including core fucosylation, on BCR signal transduction in the B cell lines studied here, they may affect mIgG-BCR function of primary B cells (e.g., resting memory B cells), especially when studied in the context of other immune cells. However, testing this would be a major technical challenge in the human system, as it is difficult to obtain and maintain antigen-specific human memory B cells in sufficient numbers to knock out Fc glycan sites and perform functional experiments.

In addition, our study sheds light on the structure of γHC glycans expressed on BCRs, which was currently, to the best of our knowledge, unknown. We report that the conserved FcG on Ramos IgG-BCRs are mainly A2 complex-type glycans carrying core fucosylation, 77–88% galactosylation, 16–21% sialylation, and 46–52% bisection. We observed a similar glycosylation profile for IgG secreted by human Ramos B cells, suggesting that mIgG and sIgG follow similar biosynthesis routes and that antibody sialylation occurs B-cell

intrinsically, as has also been evidenced by others[35]. Nonetheless, since antibody-secreting cells (i.e., plasma blasts and plasma cells) belong to different B-cell subsets compared to B cells expressing membrane-bound IgG (naïve and memory B cells), it is also conceivable that Fc glycan compositions differ between IgG-BCRs and secreted IgG[27]. As the mIgG Fc glycan profile was obtained after B cell lysis, a minor fraction of high-mannose glycans was observed. This is probably best explained by co-capturing of IgG-BCRs in the endoplasmic reticulum (ER) or *cis*-Golgi, where proteins express glycans that still need to be processed by glycosidases and glycosyltransferases during transfer to the *medial*- and *trans*-Golgi[36]. After capturing surface biotinylated IgG-BCRs, no apparent high-mannose glycans could be detected. In addition, no mannose-rich glycans were detected in the Fc domain of sIgG, as all intact immunoglobulins secreted by B cells passed the entire Golgi network.

In summary, we report that γHC glycosylation has no effect on the stable expression of functional IgG-BCRs on the surface of human B cells, implying that this conserved glycan modification plays no role in heavy and light chain assembly or the interaction with the signaling subunits Igα and Igβ. Similarly, BCRs show no difference in antigen

binding, subsequent uptake of BCR-antigen complexes, BCR signaling, or the secretion of intact IgG in the absence or presence of γHC glycans. As expected, we show that the absence of Fc glycosylation results in IgG that cannot mediate ADCC or CDC. These results indicate that γHC glycosylation likely evolved as a feature to control the effector functions of secreted IgG, but not as a mechanism to determine B-cell fate.

## Methods

### Cell lines and cell culture

Human Burkitt Lymphoma (Ramos) B-cell transfectants (anti-CCP2 2G9, anti-CCP2 3F3, and anti-TT D2) were generated as described earlier[28]. In brief, IgG-BCR sequences were obtained from single B cells of ACPA+ patients with RA. Labeled CCP2/CArgP2-streptavidin tetramers or tetanus toxoids were used for antigen-specific B-cell isolation[37]. BCR sequencing was performed from cDNA of single sorted B cells using ARTISAN-PCR as described earlier[38]. VH and VL were together with the IGHG1*03 or the IGKC constant domain (Uniprot), the Kozak sequence, and the IGHV1-18*01 leader sequence codon-optimized and ordered from GeneArt (Life Technologies). Full-length LC and γHC were cloned into the pMIG-IRES-GFP-2AP vector backbone in the presence or absence of the IGHG1 transmembrane domain using the In-Fusion HD Cloning Kit (Clontech). For the generation of FcG− variants, the γHC N(297)-linked glycosylation site was mutated into Q(297)-S-T using Site-Directed Mutagenesis PCR (NEB). Inserts were verified by Sanger sequencing performed on Applied Biosystems 96-capillary (ABI3730) systems (LGTC facility, Macrogen). Retroviral transductions were performed as described earlier[29]. Briefly, Phoenix-ECO (ATCC; CRL-3212™) cells were transfected with PolyJet DNA transfection reagent (SignaGen Laboratories). Retrovirus-containing supernatants were collected 72 hours post-transfection and used for the transduction of GFP- MDL-AID (IGHM, IGHD, IGLC, and AID) KO Ramos B cells carrying *slc7a1* (generated by Dr He, University Freiburg)[28]. B-cell lines were cultured in RPMI1640/GlutaMAX™/10% fetal calf serum (FCS)/10 mM Hepes medium (Thermo Fisher Scientific) with penicillin/streptomycin (100 U/ml; P/S) (Lonza). To compare GFP+ FcG+ and GFP+ FcG− cells in functional assays, B-cell lines were sorted by identical GFP and mIgG-BCR expression using 1 µg/ml AF647 NHS ester (Thermo Fisher; A2006) labeled Fab fragment goat anti-human IgG (Jackson ImmunoResearch; 109-007-003) on a CytoFlex SRT cell sorter (Beckman Coulter).

To analyze the ability of FcG+ and FcG− IgG1 antibodies to bind FcγRs and activate the complement system, we produced both variants of 2G9 in Freestyle 293-F cells (Gibco)[28,39]. Therefore, LC and γHC or Q(297)-mutant γHC sequences were cloned into pcDNA3.1 (+) expression vectors using the In-Fusion HD Cloning Kit (Clontech) as mentioned above.

### Flow cytometry

Membrane-bound IgG (BCR) expression of transduced FcG+ and FcG− 2G9, 3F3, and D2 GFP + B-cell lines and the non-transduced MDL-AID KO GFP- B-cell line was analyzed using flow cytometry. B cells were stained with 0.5 µg/ml goat anti-human IgG-Fc phycoerythrin (PE) (eBioscience™; 12-4998-82; lot: 2481260) in staining solution (PBS/0.5%BSA/0.02% NaN3) on ice for 30 min. To analyze BCR expression over time, rainbow beads (Spherotech; RFP-30-5A) were used for assay standardization. To determine antigen binding, B cells were stained with Allophycocyanin (APC)-labeled CCP2/CArgP2-streptavidin tetramers (0−5 µg/ml)[37] or APC-TT (0−5 µg/ml) on ice for 30 min. For B-cell lines directed against CCP2, binding to the CArgP2 negative-control antigen was subtracted.

BCR downmodulation after stimulation was assessed as described earlier[39]. In short, 0.2 million B cells were incubated for 30 min on ice followed by 15 min stimulation at 4 °C either with control PBS, CCP2-/CArgP2-streptavidin tetramers (5 µg/ml), TT-streptavidin tetramers (1.5 µg/ml) or 1 µg/ml goat F(ab')2 anti-human kappa (SouthernBiotech; 2062-01) in PBS/2% FCS. Stimulated B cells were incubated for 0, 5, 15, or 30 min at 37 °C to allow BCR-antigen uptake. The surface remaining IgG-BCRs were stained with AF647 NHS (*N*-hydroxysuccinimide) ester (Thermo Fisher Scientific; A20006) labeled Fab goat anti-human IgG (Jackson ImmunoResearch; 109-007-003) diluted 1:2000 in staining solution.

IgG-BCR activation was analyzed by the intracellular expression of phosphorylated Syk (pSyk) after antigen (0−25 µg/ml) or anti-IgG (0 to 25 µg/ml) stimulation as described earlier[39]. Briefly, 0.3 million B cells were stimulated with CCP2-streptavidin tetramer, TT-streptavidin tetramer or goat anti-human IgG F(ab')2 (Jackson ImmunoResearch; 109-006-097; lot: 164786) at 37 °C in stimulation medium (RPMI/100 U/ml P/S/GlutaMAX™/10 mM Hepes/1% FCS). The amount of the stimulus is given in the respective figure legend. Afterwards, cells were fixed (BioLegend Fixation Buffer; 420801) and permeabilized (True-Phos™ Perm Buffer; 425401). After washing, the intracellular expression of phosphorylated Syk was determined with a mouse anti-human pSyk(Y319)-AF647 mAb (17 A/P-ZAP70; BD; 557817; lot: 9165873) diluted 1:10 in staining solution. The rate of pSyk expression was calculated as the median fluorescence intensity (MFI) ratio between stimulated and unstimulated cells. Gating was based on stimulated MDL-AID KO control B cells. Stained cells were analyzed on a BD LSR-II flow cytometry instrument. Data were analyzed with FlowJo_V10.

### Calcium flux measurement

B-cell activation was determined by calcium release after anti-IgG stimulation, as described earlier[39]. In brief, 1 million B cells were stained in 200 µl calcium- indicator loading dye medium containing 2 µM Indo-1 AM (Abcam; ab142778) and 0.05% pluronic acid (Molecular Probes; P6866) in stimulation medium (RPMI1640/100 U/ml P/S/GlutaMAX™/10 mM Hepes/2%F°S) for 35 min at 37 °C in the dark. After washing, B cells were incubated with 500 µl stimulation medium plus 2 mM calcium on ice and in the dark until usage. 15 min before the flow analysis, B cells were prewarmed at 37 °C to decrease baseline activation upon measurement. The analysis was performed on a Cytek Aurora 5 L instrument including a UV laser, and acquiring 500 cells/s at a high speed. After -1.5 min of baseline measurement, PBS or 20 µg/ml goat anti-human IgG F(ab')2 (final concentration) (Jackson ImmunoResearch; 109-006-097; lot: 164786) was added and mixed adequately. The measurement continued for another 6 min until the signal reached baseline again. Calcium flux was measured as the ratio of calcium-bound Indo-1 to unbound Indo-1. Calcium flux (peak-baseline signal) and kinetics (AUC) were analyzed. Data were analyzed with FlowJo_V10.

### Isolation and gel electrophoresis of IgG-BCR and secreted IgG

To capture IgG-BCRs, 20 million Ramos B cells were washed with PBS, to remove FCS from the cell culture medium, and lysed in 8 ml PBS + 1% Triton-X100 for 60 min at 37 °C. IgG-BCRs were captured from cell lysis supernatants using 20 µl CaptureSelect™ FcXL Affinity Matrix (Thermo Fisher Scientific) and an overnight incubation at 4 °C. B cell secreted IgG were captured from 12 ml Ramos B cell supernatant (cultured at a density of two million cells/ml) by a 4 °C overnight incubation with 20 µl CaptureSelect™ FcXL Affinity Matrix (Thermo Fisher Scientific). Laemmli sample buffer (4×) (Bio Rad) was added to the IgG/FcXL bead slurry, boiled for 5 min at 95 °C, and loaded on a 4 to 15% SDS gel (Bio Rad). Proteins were detected with Coomassie Brilliant Blue G-250 Dye (Thermo Fisher Scientific) or the SilverQuest™ Silver Staining Kit (Thermo Fisher Scientific) according to the manufacturer's instructions. The size was determined using the PageRuler™ Plus Prestained Protein Ladder (Thermo Fisher Scientific).

## Surface biotinylation and membrane-IgG glycan analysis by western blot

Ramos cell surface biotinylation was performed according to the manufacturer's instructions for the Cell Surface Protein Biotinylation and Isolation Kit (Pierce; A44390). In brief, cells were biotinylated using Sulfo-NHS-SS-Biotin for 30 min at RT. After cell lysis, biotinylated IgG was captured using NeutrAvidin™ Agarose and eluted in elution buffer containing 10 mM DTT for sample reduction. For further sample denaturation, 2% SDS was added, and the samples were incubated for 30 min at 60 °C. Reduced biotinylated IgG (-10 ng) were treated with 2U EndoH (Roche) in 50 mM sodium acetate buffer pH 5.5 or 2U PNGaseF (Roche) in 1:1 5× PBS/4% NP-40 overnight at 37 °C. The presence of high-mannose (EndoH treatment) or other (PNGaseF treatment) N-glycans on the captured IgG was identified via a size-shift by Western blot analysis. Therefore, lysates were mixed with 4× Laemmli-buffer (BioRad), incubated for 5 min at 95 °C, and loaded on a 4% to 15% SDS-polyacrylamide gel (BioRad). Immunoblotting was performed on a Nitrocellulose membrane (BioRad). Blots were incubated in 3% skim milk powder/PBS/0.05% Tween (PTE) for 2 h at RT. Following washing with PBS/0.05% Tween (PT), blots were incubated at 4 °C overnight with goat anti-human IgG (Invitrogen; 31410) diluted 1:1000 in PTE. Bound antibodies were visualized using enhanced chemiluminescence (GE Healthcare; RPN-2109). Readout was performed on a BioRad Chemidoc Touch Imaging system.

## In gel IgG-Fc peptide glycan analysis by LC-MS

The IgG-BCR protein bands were extracted from the SDS gel, to exclude glycosylated contaminants and transferred to 1.5 ml Eppendorf safe-lock microcentrifuge tubes (Merck). Gel pieces were destained according to the SilverQuest™ Silver Staining Kit (Thermo Fisher Scientific), washed with 25 mM ammonium bicarbonate (ABC), followed by 100% ACN, and, for protein reduction, incubated for 30 min at 56 °C in 10 mM dithiothreitol (DTT)/25 mM ABC. After reduction, gel pieces were washed with 100% ACN and, for alkylation, incubated for 30 min at RT in the dark in 55 mM iodoacetamide/25 mM ABC to block reactive cysteine groups. After washing with 25 mM ABC and 100% ACN, gel bands were dried in a centrifugal vacuum concentrator for 5 min. In gel, proteins were digested by adding 16.5 µg/ml sequencing grade modified trypsin (Promega; V5111) in 25 mM ABC, followed by overnight incubation at 37 °C. Trypsinised IgG (1–2 µl) were separated on an Ultimate 3000 UHPLC system (Dionex/Thermo Fisher Scientific, Breda, The Netherlands) coupled to a MaXis Impact HD quadrupole-TOF mass spectrometer (MS) (MaXis HD, Bruker Daltonics, Bremen Germany) equipped with a CaptiveSpray NanoBooster source (Bruker Daltonics, Bremen, Germany)[40,41]. Briefly, trypsinised samples were extracted by a C18 trap column (Acclaim PepMap 100; 100 µm by 2 cm, particle size of 5 µm, pore size 100 Å; Dionex/Thermo Fisher Scientific) and washed for 2 min with 15 µl/min of 0.1% formic acid (FA)/1% ACN. Samples were separated on a C18 analytical liquid chromatography (LC) column (Acclaim PepMap 100; 75 µm by 15 cm, particle size of 3 µm, pore size of 100 Å; Dionex/Thermo Fisher Scientific), and elution was performed at a flow rate of 700 nl/min with buffer A [0.1% FA (v/v)] and buffer B [95% ACN/0.1% FA (v/v)]. A gradient of 1% to 70% buffer B in 70 min was applied ($t = 0$ min, B = 1%; $t = 5$ min, B = 1%; $t = 30$ min, B = 50%; $t = 31$ min, B = 70%; $t = 35$ min, B = 70%; $t = 36$ min, B = 1%; $t = 70$ min, B = 1%). The CaptiveSpray NanoBooster was operated with ACN-enriched gas (0.2 bar) and dry gas (3 liters/min) at 180 °C and a capillary voltage of 1150 V. Mass spectra were acquired within a mass range of m/z 550 to 1800. Data were collected using Compass 1.9 for OTOF version 4.0.15.3248 (Bruker Daltonik GmbH). Data were assessed using DataAnalysis (Bruker Daltonics) and the calculated masses of IgG1 Fc glycan peptides after tryptic digestion. Data processing, including peak integration, was performed using LaCyTools v1.1.0 (https://github.com/Tarskin/LaCyTools). Quality control (QC) was based on a signal-to-noise (S/

N) ratio above nine, a mass accuracy of +/−20, and an isotopic peak quality of 0.2[42]. The degree of galactosylation (G), sialylation (S), fucosylation (F), and the frequency of bisecting N-acetylglucosamine (GlcNAc, N) were calculated as described earlier[41].

## Enzyme-linked immunosorbent assay (ELISA)

To analyze the amount of functional secreted IgG by Ramos B cells, an antigen-specific IgG ELISA was performed. For this purpose, the B cell supernatant was diluted (from 1:4 to 1:64) in PBS/0.05%Tween/1% BSA (PBT) and added overnight at 4 °C on streptavidin plates (Microcoat; 65001) coated with biotinylated CCP2 (1 µg/ml). Deposition of anti-CCP2 IgG to the ELISA plate was detected with an HRP-conjugated rabbit-anti-human IgG secondary detection antibody (DAKO; P0214; lot: 20036015; 1:3000). The ELISA was developed with ABTS and $H_2O_2$, and the absorbance was read at 415 nm.

To assess classical complement activation of the secreted anti-CCP2 IgG, complement ELISA was performed as described earlier[30]. Briefly, IgG was added on CCP2-coated plates (4 µg/ml), and anti-CCP2 IgG deposition was detected as described above. To assess complement deposition, 0% or 1% complement-active normal human serum (NHS) diluted in GVB++ (veronal buffered saline containing 0.5 mM MgCl2, 2 mM CaCl2, 0.05% Tween 20, and 0.1% gelatin [pH7.5]) were added to the anti-CCP2 IgG coated ELISA plates for 1 hour at 37 °C. Complement deposition was determined with rabbit-anti-C1q (DAKO; A0136; 1:1000), goat anti-C4 (QUIDEL; A305; 1:1000), or rabbit-anti-C3c (DAKO; A0062; 1:1000) secondary antibodies, added in PBT for 1 hour at 37 °C. Binding was detected with HRP-labeled goat anti-rabbit (DAKO; P0448; 1:3000) or rabbit-anti-goat (DAKO; P0449; 1:3000) detection antibodies, each diluted in PBT at 37 °C for 1 hour. ELISA readout was performed using ABTS and $H_2O_2$ and an absorbance at 415 nm.

## SPR measurements

SPRs measurements were performed on an IBIS MX96 (IBIS technologies) as described earlier[43]. In brief, biotinylated hFcγRs were spotted using a Continuous Flow Microspotter (Wasatch Microfluidics) onto a single SensEye G-streptavidin sensor (Ssens; 1-08-04-008). The hFcγRs were spotted in dilutions ranging from 10 nM to 0.3 nM for hFcγRIIa 131H, hFcγRIIa 131 R and hFcγRIIβ, from 30 nM to 1 nM for hFcγRIIIa 158 F and hFcγRIIIβ NA2, and from 100 nM to 3 nM for hFcγRIIIa 158 V in PBS + 0.075% Tween-80 (VWR; M126−100 ml), pH 7.4. Biotinylated anti-His mIgG1 (GenScript; A00613) was spotted in duplicate and 3-fold dilution, ranging from 30 nM to 1 nM. Subsequently, 50 nM His-tagged hFcγRI was loaded onto the sensor. IgG was then injected over the IBIS at a twofold dilution series starting at 0.49 nM until 1000 nM in PBS + 0.075% Tween-80. Regeneration after every sample was performed with 10 nM Glycine-HCl, pH 2.4. The dissociation constant ($K_D$) was calculated by equilibrium fitting to $R_{max} = 500$. Association and dissociation curves of His-tagged hFcγRI were subtracted before the calculation of IgG binding affinity using SPRINT 1.9.4.4 software (IBIS technologies). Analysis and calculation of all binding data was performed with Scrubber software version 2 (Biologic Software) and Excel.

## Statistics and reproducibility

Statistical analyses were performed using GraphPad Prism. (Multiple) paired or unpaired two-sided $t$ tests were performed as indicated in the respective figure legends. The number of biologically independent replicates (B cells or antibodies were harvested and processed in individual experiments on different days) or technical replicates (samples were processed in the same experiment) are indicated in the respective figure legends. No statistical method was used to predetermine sample size. No data were excluded from the analyses. The experiments were not randomized. The Investigators were not blinded to allocation during experiments and outcome assessment.

**Reporting summary**

Further information on research design is available in the Nature Portfolio Reporting Summary linked to this article.

## Data availability

We confirm that the data supporting the findings of this study are available within the article and/or can be provided upon request. The processed flow cytometry and LC-MS data are provided in the Source Data file. The LC-MS raw data have been deposited to the ProteomeXchange Consortium via the PRIDE[44] partner repository with the dataset identifier PXD047089. The Ramos MDL-AID KO cells can be provided by Prof. Dr. Reth (Albert-Ludwigs-Universität Freiburg, Germany). Source data are provided with this paper.

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

## Acknowledgements

The authors thank J.-W. Drijfhout (LUMC, Leiden, The Netherlands) for providing the CCP2/CArgP2 peptides. This work was supported by ReumaNederland LLP5 and 17-1-402 (to R.E.M.T.), ZonMw TOP 91214031 (to R.E.M.T.), Target-to-B LSHM18055-SGF (to R.E.M.T.), the European Research Council (ERC) advanced grant AdG2019-884796 (to R.E.M.T.) and the ERC GlycanSwitch grant 101071386 (to M.W.). Views and opinions expressed are, however, those of the author(s) only and do not necessarily reflect those of the European Union or the ERC Executive Agency. Neither the European Union nor the granting authority can be held responsible for them.

## Author contributions

All authors drafted the article and revised it critically for important intellectual content, and all authors approved the final version to be published. T.K., D.v.d.W., M.W., G.V., and R.E.M.T. designed research; T.K., V.F.A.M.D., A.E.H.B., C.K., and L.H. performed research; T.K., V.F.A.M.D., and A.E.H.B. analyzed data; and T.K. and R.E.M.T. wrote the original draft of the article.

## Competing interests

The authors declare no competing interests.
