## [Peer Review File · Nature Communications]

N-linked Fc glycosylation is not required for IgG-B-cell receptor function in a GC-derived B-cell lineREVIEWER COMMENTS

Reviewer #1 (Remarks to the Author):

In this interesting manuscript Kissel et al. address the question of whether membrane bound IgG has similar Fc glycosylation as secreted IgG and if that Fc glycosylation impacts the function of membrane bound IgG BCR. They engineered Ramos B cell lines to express BCRs with known specificities in which the N297 residue has been mutated to prevent glycosylation or was left unmutated. Their main finding is that, within the limitations of a cell line in culture, they find no impact of Fc glycosylation on IgG BCR expression or function. This contrasts with soluble IgG, suggesting that IgG Fc glycosylation only controls effector function of the secreted IgG. It is a well written manuscript. Given the growing appreciation of the impact of immunoglobulin glycosylation in immunoglobulin function it should be of interest to the community.

Specific comments and questions:

It is unclear from the material and methods and text how the cell lines that are shown were generated. There are no GFP negative cells (Fig S1), so were these cells sorted for GFP+ and if so, were they selected for B cells with equal GFP expression, with the assumption that that reflects equal IgG BCR expression? This is relevant for the question of the potential effects of glycosylation on surface IgG expression, if there was a selection that is not shown.

It would be nice to see some statistical analysis of replicates in Fig 1B. Equal IgG BCR expression is also relevant variable when analyzing BCR stimulation later in the manuscript.

In the analysis of glycosylation of the Fc peptide of the IgG BCR, cells (Fig Fig 1 D/E) were lysed and IgG was captured with CaptureSelect FcXL. As acknowledged by the authors that approach would capture both surface IgG BCR and any intracellular IgG that may still be glycan maturing in the ER-Golgi. How big a contamination is that? For analysis of secreted IgG that is obviously not a problem but, in this case, would surface biotinylation to mark the IgG BCRs prior to cell lysis and then a sequential immunoprecipitation have not been a better approach?

I would have predicted an effect on signaling due to recruitment of inhibitory receptors. The data argues differently. The authors did a good job discussing limitations of the system that may have missed scenarios where there could have been an impact of Fc glycosylation.

To help interpret the signaling data shown it would be nice to see the surface IgG expression of the different experiments to exclude that variable or, even better, use a non-stimulatory fluorescently-labeled anti-IgG Fab fragment to allow to gate on cells with equal IgG BCR expression and then compare Syk/Ca²⁺ responses.

Only one time point is shown for Syk, but the calcium traces give some insight into the kinetics. It would be useful if the authors would reanalyze the calcium data shown and plot the area under the curve of the different experiments. It looks in the plots shown that the 2G2 cell line and the D2 cell line may be slower to return to baseline in the absence of Fc glycosylation, which may reflect a lack of negative feedback signaling. This is missed if you measure peak signal.

Can the membrane-bound IgG activate complement when glycosylated (i.e. if you add human serum to the cell cultures)? C3 deposition might alter signaling through CD21-CD19.

Reviewer #2 (Remarks to the Author):

Kissel et al describe in the manuscript "Fc glycosylation is not required for IgG-B-cell receptor function" very important results demonstrating that the loss of γ -heavy chain (γ HC) N-glycans has no effect on several functions of IgG-B-cell receptor (BCR). They used three different Burkitt Lymphoma (Ramos) B-cell systems: 1) B cells without BCR and activation-induced cytidine

deaminase (AID), 2) B cells transduced with GFP and mIgG-BCR and 3) B cells transduced with GFP and mIgG-BCR without the γ HC N-glycosylation site. They observed in a number of elegant experiments that the loss of γ HC N-glycans has no effect on the expression-levels of IgG-BCR, the antigen binding of BCR, the BCR downmodulation and BCR- signaling. In contrast, released IgG without Fc-N-glycans showed a disturbed mechanism of action as described before by many researchers. In addition, the structure of γ HC N-glycans were characterized. The study is well written (including the material and methods section) and very interesting. However, additional experiments are suggested to proof some of the hypotheses:

- The title of the manuscript is "Fc glycosylation is not required for IgG-B-cell receptor function". Can the authors exclude that O-glycans are presents on IgG-BCR (e.g. close to the transmembrane domain) and functional?

- The results contradict previous analyses demonstrating that core fucosylation of γ HC N-glycans is needed for BCR function (Li et al. 2015 J Immunol). Kissel et al criticize the used experimental design, since in Li et al a loss of core-fucosylation on all N-glycans was induced. Thus, a BCR-specific explanation is not possible. This sounds plausible. Since Kissel et al has 3 powerful cell-systems, experiments with fucosyltransferase-inhibitors (e.g. peracetylated 6-alkynyl-fucose (6-Alk-Fuc) and 2-deoxy-2-fluorofucose (2FF) elegant) are suggested to proof if a complete loss of core-fucosylation leads to the describes phenotype of Li et al in the used cell systems.

- The authors observed high-mannose glycans on IgG-BCR and speculate that these N-glycans are from IgG-BCR in the ER and/or Golgi (not completely processed), since they used cell lysates. They can easily proof their suggestion, since several effective methods are available to isolate only membrane proteins (e.g. sulfo-NHS-SS-biotin).

- The authors calculate the percentage of Fc peptide glycans. How robust are these values (n=?, Standard deviation,...). The authors should show how reproducible the result are.

Reviewer #3 (Remarks to the Author):

In this study, Toes et al. made an AID-BCR null Ramos (GC-like) cell line and knocked in physiological IgG BCR recognizing tetanus toxoid or citrullinated proteins (autoantigens), with (N297) or without (Q297) glycosylation potential. In the N297 lines, they show that the BCR is glycosylated as expected. In functional studies, IgG BCR internalization and activation of a downstream signalling molecule (phospho-Syk) were tested and glycosylation shown not to impact consequences of antigen exposure to the B cell-stage, but to be important for function of the antibodies. This latter part is expected from the literature, but this is the first work in my knowledge to test functional consequences of glycosylation on BCR signalling events.

Major points:

1) Glycosylation and glycan function differs for B cells in different immune response stages, specifically differences in naïve and GC B cell states (eg PMID: 30564237). The study title should reflect that this is specific to IgG-BCR function in a GC-like B cell line (Ramos), rather than primary B cells. Further, the abstract reads as if the study were in primary B cells, so the abstract should reflect the study is in cell lines too.

2) L155 and Fig 3: While glycosylation may not affect signalling maxima, it may shift affinity-based thresholding. Fig 3A would be stronger if a titration of antigen/anti-IgG were used to show the dose-response of the lines to the stimulation. Similarly, a timecourse of pSyk expression in at least one FcG + and one neg cell line would be helpful to know that stimulation follows the normal kinetics of shutdown.

3) Although probably beyond the scope of the study, it would be interesting to know by superresolution microscopy if the nanoscale compartmentalization of IgG on the surface is affected by Fc glycosylation/fucosylation, as that could provide some indication of activation potential being normal as well.

Minor points:

1) L49: Under normal conditions (i.e. non-transformed cell lines) Igs are produced by plasmablasts

and plasma cells which differentiate from B cells. Please adjust terminology.

2) L223: Ref 23 is a key study that precedes this work. Although the study does not perform assays in a BCR-targeted way as done here, one could contend that the study shows that core N-fucosylation is required for oligomerization or relocalization of the IgG BCR, without showing that it necessarily impacts signalling under the limited circumstances tested in the study by Toes et al. Given the lack of titration and kinetic experiments in Fig 3, some elaboration on this nuanced possibility is warranted in the discussion.

3) Ramos is a GC-like cell line, so this study may be relevant for GC Stage B cells rather than those in the resting memory state. This potential state limitation should be acknowledged.

4) Fig 1-4 it is ambiguous how many times experiments were performed. Does 'independent replicates' mean technical replicates or experiments performed with n=1 on different days? Please clarify.

5) Fig 3C anti-CCP2 studies have only two replicates, which seems underpowered to reveal any potential difference between treatments, despite the responses looking highly similar. If I am correct in thinking that the assays are designed to compile data from separate experiments for each symbol shown, I suggest this assay should be repeated at least one more time to strengthen the power of the statistical test.

6) Figure 4: L615. Please state that this is from technical replicates from one experiment.

7) Figure 2C: TT treatments are from two replicates only, so it isn't really valid to assess this distribution with a statistical test. I suggest the assay should be repeated at least a third time.

8) L120: positive for positive

9) L197: affinity for affinity

10) L585: shown for shown

Reviewer #4 (Remarks to the Author):

This is an interesting study in which the authors examine the HC glycosylation of IgG BCRs on B cells. They do this by expressing IgG BCRs with or without glycosylation site on the Fc part on a human B cell line which lacks endogenous BCRs. They show that the transduced IgG BCRs have a similar complex glycosylation pattern as IgM BCRs would have. They found no effect of this glycosylation on Ag binding, signaling or BCR internalization on these IgG BCRs. Then they also express secreted versions of these IgGs and find the expected strong differences in effector functions of the secreted IgG in Fc receptor binding and complement binding. Although the results are largely "negative", it is an important study that demonstrates that the different effector functions of IgGs that carry differential glycosylation patterns can just be explained by the secreted antibodies themselves, but not by a differential function or selection of IgG BCRs on memory B cells.

The study is well performed and the experiments are very well done, shown and described. No modifications of experiments are necessary, to my opinion.

In the introduction the authors describe the structure of the IgG BCRs, the association to Ig-alpha and Ig-beta. This description lacks the differences to the IgM-BCR. The membrane bound IgG has a long intracellular tail, containing an ITT motif which is important to enhance IgG signaling, compared to IgM signaling. See Engels et al. Nat Commun 2014 and Wienands and Engels Immunol Lett 2016. This presence of the ITT motif in IgG BCRs may also be important for the discussion in this manuscript, because it may explain Ig-alpha and Ig-beta independent signal functions.

Point by point response to reviewers' comments

Reviewer #1 (Remarks to the Author):

In this interesting manuscript Kissel et al. address the question of whether membrane bound IgG has similar Fc glycosylation as secreted IgG and if that Fc glycosylation impacts the function of membrane bound IgG BCR. They engineered Ramos B cell lines to express BCRs with known specificities in which the N297 residue has been mutated to prevent glycosylation or was left unmutated. Their main finding is that, within the limitations of a cell line in culture, they find no impact of Fc glycosylation on IgG BCR expression or function. This contrasts with soluble IgG, suggesting that IgG Fc glycosylation only controls effector function of the secreted IgG. It is a well written manuscript. Given the growing appreciation of the impact of immunoglobulin glycosylation in immunoglobulin function it should be of interest to the community.

Specific comments and questions:

1. It is unclear from the material and methods and text how the cell lines that are shown were generated. There are no GFP negative cells (Fig S1), so were these cells sorted for GFP+ and if so, were they selected for B cells with equal GFP expression, with the assumption that that reflects equal IgG BCR expression? This is relevant for the question of the potential effects of glycosylation on surface IgG expression, if there was a selection that is not shown.

Response:

We thank the reviewer for the relevant comments and have revised the manuscript accordingly.

BCR- and GFP-transduced B cells carrying mIgG with and without Fc glycans were sorted by identical BCR and GFP expression before performing the functional experiments. Detailed information on the generation of the cell lines has now been included in the material and method section (lines 550 – 554). In addition, we have included the dot-plot flow data of the non-transduced GFP and mIgG (BCR) negative MDL-AID KO cells to the revised Supplementary Fig. 1a. To analyze the potential effect of Fc glycosylation on mIgG expression over time, we have cultured the cell lines for 20 days and identified mIgG (BCR) expression over culture time. These data have now been included to the revised manuscript (Fig. 1d, Supplementary Fig. 1b) and show no differences in expression over time of BCRs that do- or do not express Fc-glycans.

2. It would be nice to see some statistical analysis of replicates in Fig 1B. Equal IgG BCR expression is also relevant variable when analyzing BCR stimulation later in the manuscript.

Response: In response to the reviewer's comment, we have now included a figure (Fig. 1c) showing 7 biological replicates of the mIgG-BCR staining and the corresponding statistical analyses. In addition, we analyzed mIgG-BCR expression over time to determine whether Fc glycans affect BCR stability and thus could affect functional experiments after several days of culturing. As indicated above, these additional results are now shown in Fig. 1d and Supplementary Fig. 1b.

3. In the analysis of glycosylation of the Fc peptide of the IgG BCR, cells (Fig Fig 1 D/E) were lysed and IgG was captured with CaptureSelect FcXL. As acknowledged by the authors that approach would capture both surface IgG BCR and any intracellular IgG that may still be glycan maturing in the ER-Golgi. How big a contamination is that? For analysis of secreted IgG that is obviously not a problem but, in this case, would surface biotinylation to mark the IgG BCRs prior to cell lysis and then a sequential immunoprecipitation have not been a better approach?

Response: We agree with the reviewer that surface biotinylation and subsequent capture of biotinylated IgG would be a good approach to specifically isolate and analyze membrane-bound IgG. Unfortunately, this approach did not allow us to obtain sufficient material for subsequent Fc peptide glycan analysis by LC-MS. Based on our data we estimate that at least 3% of the obtained BCRs are non-cell surface derived. These BCRs present high-mannose glycans, which are known to have not yet fully matured into complex-type glycans as they have not been processed in the ER or Golgi. This percentage is based on the fraction of the high-mannose cluster relative to the total Fc glycans observed (Fig. 1f). In response to the reviewer's comment, we now further confirmed that mIgG predominantly express complex-type N-glycans by Western blot analysis of biotinylated surface IgG followed by treatment with EndoH (cleaves mannosylated glycans) or PNGaseF (cleaves all N-linked glycans), a method previously described by others¹. The results show only a size shift of mIgG heavy chains after PNGaseF treatment without an apparent effect by EndoH treatment. These additional WB data have now been included to the revised version of the manuscript (Supplementary Fig. 2a, material and methods lines 616 – 6232 and results lines 141 - 145).

4. I would have predicted an effect on signaling due to recruitment of inhibitory receptors. The data argues differently. The authors did a good job discussing limitations of the system that may have missed scenarios where there could have been an impact of Fc glycosylation.

To help interpret the signaling data shown it would be nice to see the surface IgG expression of the different experiments to exclude that variable or, even better, use a non-stimulatory fluorescently-labeled anti-IgG Fab fragment to allow to gate on cells with equal IgG BCR expression and then compare Syk/Ca²⁺ responses.

Response: We thank the reviewer for highlighting this issue. We have now included data showing mIgG-BCR expression over time, within the time frame in which the cells were used for functional experiments (20 days), to the revised version of the manuscript. As shown in Fig. 1d and Supplementary Fig. 1b, mIgG-BCR expression, and in particular the ratio between the FcG⁺ and FcG⁻ cell lines, is stable over time. Thus, mIgG expression can likely be excluded as an additional variable affecting subsequent functional responses. This is now specifically mentioned in the revised manuscript (lines 241 – 243). In addition, we have now reanalyzed the calcium-flux data gating on identical GFP expression between FcG⁺ and FcG⁻ cell lines as a proxy for identical BCR expression. These data are now shown in the revised Fig. 3, e-g. We did not include an additional Fab anti-IgG staining in the functional experiments as we observed binding competition between the surface staining and the anti-IgG stimulus in previous experiments. However, to specifically address the reviewer's comment, we determined mIgG-BCR expression and calcium-flux AUCs within the same experiment and show these additional data in Supplementary Fig. 3c.

5. Only one time point is shown for Syk, but the calcium traces give some insight into the kinetics. It would be useful if the authors would reanalyze the calcium data shown and plot the area under the curve of the different experiments. It looks in the plots shown that the 2G2 cell line and the D2 cell line may be slower to return to baseline in the absence of Fc glycosylation, which may reflect a lack of negative feedback signaling. This is missed if you measure peak signal.

Response: In response to the reviewer's comment, we reanalyzed the calcium flux data in the revised version of the manuscript. In addition, we repeated the calcium flux experiments for all B-cell lines and performed statistical analyses. The additional calcium flux kinetic data (area under the curve) are now shown in Fig. 3f. We found no significant differences in calcium flux peak and kinetics between FcG⁺ and FcG⁻ B cell lines.

6. Can the membrane-bound IgG activate complement when glycosylated (i.e. if you add human serum to the cell cultures)? C3 deposition might alter signaling through CD21-CD19.

Response: We thank the reviewer for the interesting question, but feel that these additional experiments are beyond the scope of this manuscript. To the best of our knowledge, complement activation has so far only been suggested for IgM-BCRs in transgenic mouse models expressing membrane-bound IgM deficient in C1q-binding². We could not find literature reporting complement activation of membrane-bound IgG on human B cells. Therefore, sorting out this mechanism will provide a novel pathway by which mIgG can be activated. Convincing evidence of this mechanism, if present, will likely require a considerable amount of work and time, which we feel would be beyond the scope of the current manuscript which focusses on the role of Fc glycans on BCR biology.

Reviewer #2 (Remarks to the Author):

Kissel et al describe in the manuscript “Fc glycosylation is not required for IgG-B-cell receptor function” very important results demonstrating that the loss of γ -heavy chain (γ HC) N-glycans has no effect on several functions of IgG-B-cell receptor (BCR). They used three different Burkitt Lymphoma (Ramos) B-cell systems: 1) B cells without BCR and activation-induced cytidine deaminase (AID), 2) B cells transduced with GFP and mIgG-BCR and 3) B cells transduced with GFP and mIgG-BCR without the γ HC N-glycosylation site. They observed in a number of elegant experiments that the loss of γ HC N-glycans has no effect on the expression-levels of IgG-BCR, the antigen binding of BCR, the BCR downmodulation and BCR- signaling. In contrast, released IgG without Fc-N-glycans showed a disturbed mechanism of action as described before by many researchers. In addition, the structure of γ HC N-glycans were characterized.

The study is well written (including the material and methods section) and very interesting. However, additional experiments are suggested to proof some of the hypotheses:

1. The title of the manuscript is “Fc glycosylation is not required for IgG-B-cell receptor function”. Can the authors exclude that O-glycans are presents on IgG-BCR (e.g. close to the transmembrane domain) and functional?

Response: We thank the reviewer for the valuable comments.

Here, we generated B cell lines expressing membrane-bound IgG1 B cell receptors containing only one conserved N(297)-linked glycan site in the CH2 domain. Additional O-linked glycans have only been reported for the hinge region of the IgG3 subclass³, but not for the Fc domain of IgG1 molecules. Although we cannot rule out the possibility that additional O-linked glycans on the Fc domain play a role, since their presence was not analyzed, we consider this an unlikely option since IgG1 Fc domains have not been shown to contain O-glycans. This information has now been included in the revised version of the manuscript (lines 51 - 54). In addition, in response to the reviewer's comments, we have changed the title of the revised manuscript to “N-linked Fc glycosylation is not required for IgG B-cell receptor function in a GC-derived B-cell line” for clarity and to more accurately capture the content of the manuscript.

2. The results contradict previous analyses demonstrating that core fucosylation of γ HC N-glycans is needed for BCR function (Li et al. 2015 J Immunol). Kissel et al criticize the used experimental design, since in Li et al a loss of core-fucosylation on all N-glycans was induced. Thus, a BCR-specific explanation is not possible. This sounds plausible. Since Kissel et al has

3 powerful cell-systems, experiments with fucosyltransferase-inhibitors (e.g. peracetylated 6-alkynyl-fucose (6-Alk-Fuc) and 2-deoxy-2-fluorofucose (2FF)elegant) are suggested to proof if a complete loss of core-fucosylation leads to the describes phenotype of Li et al in the used cell systems.

Response: We appreciate the feedback from the reviewer and agree that these additional experiments could be performed to confirm the findings of Li et al. However, we feel that this would address a different research question that is beyond the scope of this manuscript because the use of the indicated inhibitors does not specifically inhibit fucosylation of Fc glycans in the BCR. This manuscript aims to determine whether γ HC BCR glycans affect B cell function. Since we did not observe any effects when γ HC glycans are completely absent, we assume that no effects on BCR function are observed in the absence of BCR N-glycan core-fucosylation only.

3. The authors observed high-mannose glycans on IgG-BCR and speculate that these N-glycans are from IgG-BCR in the ER and/or Golgi (not completely processed), since they used cell lysates. They can easily proof their suggestion, since several effective methods are available to isolate only membrane proteins (e.g. sulfo-NHS-SS-biotin).

Response: In response to the reviewers' comment, we have repeated the BCR glycan analysis after cell surface biotinylation and capturing of membrane-bound IgG molecules. As indicated above, we could only perform a glycan analysis by glycosidase treatment and subsequent Western blot analysis as we had not sufficient material for the Fc peptide glycan analysis by LC-MS (please see also our response to R#1, Q#3). However, we performed additional experiments addressing this comment. The Western blot data (Supplementary Fig. 2a, material and methods lines 616 – 632 and results lines 141 – 145) show only a size-shift (N-glycan cleavage) of the γ HC after PNGaseF treatment and no apparent susceptibility to EndoH treatment (cleaves high-mannose glycans). Thus, these results indicate that mIgG-BCRs express complex-type N-glycans in the γ HC.

4. The authors calculate the percentage of Fc peptide glycans. How robust are these values (n=?, Standard deviation,...). The authors should show how reproducible the result are.

Response: In response to the reviewer's comment, we have repeated the Fc peptide glycan analyses by LC-MS (n = 4) and included the additional data to the revised Fig. 1g and h, Fig. 4e and Supplementary Fig. 2b and c. Information about the number of replicates have been added to the respective figure legends. Our data reproducibly show that mIgG BCR Fc glycans are mainly complex-type glycans containing a core fucose (100%), 77% (mIgG 2G9), 88% (mIgG 3F3) or 78% (mIgG D2) galactosylation, 47% (mIgG 2G9), 52% (mIgG 3F3) or 46% (mIgG D2) bisecting N-acetylglucosamines and 16%, (mIgG 2G9), 21% (mIgG 3F3) or 19% (mIgG D2) terminal sialic acids. Similarly, Fc glycans on sIgG (2G9) show 100% core fucosylation, 79% galactosylation, 39% bisection and 12% terminal sialic acids, being mainly S0 or S1 glycans.

Reviewer #3 (Remarks to the Author):

In this study, Toes et al. made an AID-BCR null Ramos (GC-like) cell line and knocked in physiological IgG BCR recognizing tetanus toxoid or citrullinated proteins (autoantigens), with (N297) or without (Q297) glycosylation potential. In the N297 lines, they show that the BCR is glycosylated as expected. In functional studies, IgG BCR internalization and activation of a downstream signalling molecule (phospho-Syk) were tested and glycosylation shown not to

impact consequences of antigen exposure to the B cell-stage, but to be important for function of the antibodies. This latter part is expected from the literature, but this is the first work in my knowledge to test functional consequences of glycosylation on BCR signalling events.

Major points:

1. Glycosylation and glycan function differs for B cells in different immune response stages, specifically differences in naïve and GC B cell states (eg PMID: 30564237). The study title should reflect that this is specific to IgG-BCR function in a GC-like B cell line (Ramos), rather than primary B cells. Further, the abstract reads as if the study were in primary B cells, so the abstract should reflect the study is in cell lines too.

Response: We thank the reviewer for the valuable comments.

In response to the reviewer's comment, we have adjusted the study title and abstract accordingly.

2. L155 and Fig 3: While glycosylation may not affect signalling maxima, it may shift affinity-based thresholding. Fig 3A would be stronger if a titration of antigen/anti-IgG were used to show the dose-response of the lines to the stimulation. Similarly, a timecourse of pSyk expression in at least one FcG + and one neg cell line would be helpful to know that stimulation follows the normal kinetics of shutdown.

Response: In response to the reviewer's comment, we have now included additional titration (antigen and anti-IgG) and time course (2 min to 15 min) pSyk (activation) experiments to the revised version of the manuscript. These additional data are now shown in Fig. 3c and d. In addition, we have reanalyzed the calcium flux data and also included the kinetics (area under the curve) next to the signaling maxima (flux peak) to the revised version of the manuscript (Fig. 3g). We did not observe significant differences between the FcG+ and FcG- B-cell lines.

3. Although probably beyond the scope of the study, it would be interesting to know by superresolution microscopy if the nanoscale compartmentalization of IgG on the surface is affected by Fc glycosylation/fucosylation, as that could provide some indication of activation potential being normal as well.

Response: We thank the reviewer for the comment and agree that it would be highly interesting to investigate, whether mIgG Fc glycosylation affects the spatial distribution of BCRs in a resting and activated state. Like the reviewer, we feel that acquiring these additional data with superresolution microscopy would be beyond the scope of this study. In response to the reviewer's comment, the limitation that we are not investigating the spatial organization of mIgG-BCRs has been included to the discussion section of the manuscript (lines 272 – 274).

Minor points:

1. L49: Under normal conditions (i.e. non-transformed cell lines) Igs are produced by plasmablasts and plasma cells which differentiate from B cells. Please adjust terminology.

Response: In response to the reviewer's comment, we have adjusted the terminology accordingly (lines 48 – 49).

2. L223: Ref 23 is a key study that precedes this work. Although the study does not perform assays in a BCR-targeted way as done here, one could contend that the study shows that core N-fucosylation is required for oligomerization or relocalization of the IgG BCR, without showing that it necessarily impacts signalling under the limited circumstances tested in the study by

Toes et al. Given the lack of titration and kinetic experiments in Fig 3, some elaboration on this nuanced possibility is warranted in the discussion.

Response: We agree with the suggestion of the reviewer and have included a possible contribution of mIgG Fc glycan core fucosylation on the spatial organization of BCRs to the revised version of the manuscript (lines 272 - 274). In addition, we have now included kinetic (titration and time course) experiments to the revised version of the manuscript (Fig. 3c, d and g).

3. Ramos is a GC-like cell line, so this study may be relevant for GC Stage B cells rather than those in the resting memory state. This potential state limitation should be acknowledged.

Response: In response to the reviewer's comment, we have included this limitation to the discussion of the revised version of the manuscript (lines 274 - 277).

4. Fig 1-4 it is ambiguous how many times experiments were performed. Does 'independent replicates' mean technical replicates or experiments performed with n=1 on different days? Please clarify.

Response: We apologize for this unclarity and have adjusted the figure legends of the revised manuscript accordingly. Further information on biological and technical replicates are now given in the materials and methods section (lines 699 – 703).

5. Fig 3C anti-CCP2 studies have only two replicates, which seems underpowered to reveal any potential difference between treatments, despite the responses looking highly similar. If I am correct in thinking that the assays are designed to compile data from separate experiments for each symbol shown, I suggest this assay should be repeated at least one more time to strengthen the power of the statistical test.

Response: In response to the reviewer's comment, we have repeated the calcium flux experiments for all B cell lines (revised Fig. 3e – g). We have performed paired analyses between the respective FcG+ and FcG- B cell lines measured within one experiment. In total 5 biological replicates (experiments performed on different days) are shown.

6. Figure 4: L615. Please state that this is from technical replicates from one experiment.

Response: We have adjusted the figure legend of the revised version of the manuscript accordingly.

7. Figure 2C: TT treatments are from two replicates only, so it isn't really valid to assess this distribution with a statistical test. I suggest the assay should be repeated at least a third time.

Response: In response to the reviewer's comment, we have repeated the assay (TT treatments) and included the additional data to the revised version of the manuscript (Fig. 2c).

8. L120: positive for positive

Response: We adjusted the revised version of the manuscript accordingly.

9. L197: affinity for affine

Response: We adjusted the revised version of the manuscript accordingly.

10. L585: shown for shwon

Response: We adjusted the revised version of the manuscript accordingly.

Reviewer #4 (Remarks to the Author):

This is an interesting study in which the authors examine the HC glycosylation of IgG BCRs on B cells. They do this by expressing IgG BCRs with or without glycosylation site on the Fc part on a human B cell line which lacks endogenous BCRs. They show that the transduced IgG BCRs have a similar complex glycosylation pattern as IgM BCRs would have. They found no effect of this glycosylation on Ag binding, signaling or BCR internalization on these IgG BCRs. Then they also express secreted versions of these IgGs and find the expected strong differences in effector functions of the secreted IgG in Fc receptor binding and complement binding. Although the results are largely "negative", it is an important study that demonstrates that the different effector functions of IgGs that carry differential glycosylation patterns can just be explained by the secreted antibodies themselves, but not by a differential function or selection of IgG BCRs on memory B cells.

The study is well performed and the experiments are very well done, shown and described. No modifications of experiments are necessary, to my opinion.

1. In the introduction the authors describe the structure of the IgG BCRs, the association to Ig-alpha and Ig-beta. This description lacks the differences to the IgM-BCR. The membrane bound IgG has a long intracellular tail, containing an ITT motif which is important to enhance IgG signaling, compared to IgM signaling. See Engels et al. *Nat Commun* 2014 and Wienands and Engels *Immunol Lett* 2016. This presence of the ITT motif in IgG BCRs may also be important for the discussion in this manuscript, because it may explain Ig-alpha and Ig-beta independent signal functions.

Response: We would like to thank the reviewer for the feedback. In response to the reviewer's comment, we have now included a description of the differences between membrane-bound IgG and IgM BCR assembly, positioning and glycosylation to the revised version of the manuscript (introduction lines 94 - 101). In addition, we have included the possibility of the Ig α /Ig β independent signal function of mIgG to the discussion section of the revised manuscript (lines 263 – 265).

References:

- 1 Chiodin, G. *et al.* Insertion of atypical glycans into the tumor antigen-binding site identifies DLBCLs with distinct origin and behavior. *Blood* **138**, 1570-1582 (2021). <https://doi.org/10.1182/blood.2021012052>
- 2 Roszbacher, J. & Shlomchik, M. J. The B cell receptor itself can activate complement to provide the complement receptor 1/2 ligand required to enhance B cell immune responses in vivo. *J Exp Med* **198**, 591-602 (2003). <https://doi.org/10.1084/jem.20022042>
- 3 Plomp, R. *et al.* Hinge-Region O-Glycosylation of Human Immunoglobulin G3 (IgG3). *Mol Cell Proteomics* **14**, 1373-1384 (2015). <https://doi.org/10.1074/mcp.M114.047381>

REVIEWERS' COMMENTS

Reviewer #1 (Remarks to the Author):

The authors did a good job addressing the concerns raised.

Reviewer #2 (Remarks to the Author):

The revised manuscript "Fc glycosylation is not required for IgG-B-cell receptor function" of Kissel et al was significantly improved. For instance, the authors have changed the title to "N-linked Fc glycosylation is not required for IgG B-cell receptor function in a GC-derived B-cell line" to clarify that the N-glycosylation status was characterized. In addition, they added experiments demonstrating that mIgG-BCRs contain only complex-type N-glycans. Moreover, the Fc peptide glycan analyses by LC-MS was repeated (n=4) to support the obtained results. Together with the changes, based on suggestions from the other reviewers, the authors' hypothesis was further strengthened; Very interesting results! Thus, no additional experiments or modifications are necessary.

Reviewer #3 (Remarks to the Author):

The revised manuscript adequately addresses my concerns. No further changes are suggested.

Reviewer #4 (Remarks to the Author):

All my concerns were addressed. Thank you.

Point by point response to reviewers' comments

Reviewer #1 (Remarks to the Author):

1. The authors did a good job addressing the concerns raised.

Response: We would like to thank the reviewer for the positive comment.

Reviewer #2 (Remarks to the Author):

1. The revised manuscript "Fc glycosylation is not required for IgG-B-cell receptor function" of Kissel et al was significantly improved. For instance, the authors have changed the title to "N-linked Fc glycosylation is not required for IgG B-cell receptor function in a GC-derived B-cell line" to clarify that the N-glycosylation status was characterized. In addition, they added experiments demonstrating that mIgG-BCRs contain only complex-type N-glycans. Moreover, the Fc peptide glycan analyses by LC-MS was repeated (n=4) to support the obtained results. Together with the changes, based on suggestions from the other reviewers, the authors' hypothesis was further strengthened; Very interesting results! Thus, no additional experiments or modifications are necessary.

Response: We would like to thank the reviewer for the positive feedback.

Reviewer #3 (Remarks to the Author):

1. The revised manuscript adequately addresses my concerns. No further changes are suggested.

Response: We would like to thank the reviewer for the positive comment.

Reviewer #4 (Remarks to the Author):

1. All my concerns were addressed. Thank you.

Response: We would like to thank the reviewer for the positive comment.